# Three distinct Holocene intervals of stalagmite deposition and non-deposition revealed in NW Madagascar, and their paleoclimate inferences

Voarintsoa, Ny Riavo G.[1*], L. Bruce Railsback[1], George A. Brook[2], Lixin Wang[2], Gayatri Kathayat[3], Hai Cheng[3,4], Xianglei Li[3], R. Lawrence Edwards[4], Rakotondrazafy Amos Fety Michel[5], Madison Razanatseheno Marie Olga[5]

[1] Department of Geology, University of Georgia, Athens, GA 30602-2501 U.S.A.
[2] Department of Geography, University of Georgia, Athens, Georgia, 30602-2502 U.S.A.
[3] Institute of Global Environmental Change, Xi'an Jiaotong University, Xi'an, Shaanxi 710049, P.R. China
[4] Department of Earth Sciences, University of Minnesota, Minneapolis, Minnesota 55455, U.S.A.
[5] Department of Geology, University of Antananarivo, Madagascar

*Correspondence to: Ny Riavo Voarintsoa (nv1@uga.edu or nyriavony@gmail.com)

ABSTRACT

Petrographic features, mineralogy, and stable isotopes from two stalagmites collected from Anjohibe and Anjokipoty caves allow distinction of three intervals of the Holocene in NW Madagascar. The Malagasy early Holocene (between c. 9.8 and 7.8 ka) and late Holocene (after c. 1.6 ka) intervals (MEHI and MLHI, respectively) record evidence of stalagmite deposition. The Malagasy middle Holocene interval (MMHI, between c. 7.8 ka and 1.6 ka), however, is marked by a depositional hiatus lasting for c. 6500 years.

Deposition of Stalagmites ANJB-2 and MAJ-5 from Anjohibe and Anjokipoty caves, respectively, during the MEHI and the MLHI suggests that these caves were sufficiently supplied with water to allow stalagmite formation. These MEHI and MLHI intervals may have been comparatively wet. In contrast, the long-term depositional hiatus likely suggests that the MMHI was relatively drier than the MEHI and the MLHI. This dry condition could have influenced the amount of water supplied to the cave, and thus prevented formation of the stalagmites.

The alternating "wet/dry/wet" during each of these Holocene intervals could be generally linked to the long-term migration of the Inter-Tropical Convergence Zone (ITCZ). When the ITCZ's mean position is farther south, NW Madagascar experiences wetter conditions, such as during the MEHI and MLHI, and when it moves north, NW Madagascar climate becomes drier, such as

during the MMHI. A similar wet/dry/wet succession during the Holocene has been reported in neighboring locations, such as southeastern Africa.

Stable isotope records also suggest that although the MEHI and MLHI were wetter, the stronger correlation between $\delta^{18}O$ and $\delta^{13}C$ suggest that the early Holocene vegetation closely responded to changes in climate. In contrast, the weaker correlation between $\delta^{18}O$ and $\delta^{13}C$ and the positive shift in $\delta^{13}C$ suggest that the late Holocene vegetation was controlled by something other than climate, and the plausible explanation for such changes is the practice of swidden agriculture, as reported in previous literature.

Beyond these three subdivisions, the evidence of the 8.2 ka event in the stalagmite records also suggests that climate in Madagascar was sensitive to abrupt climate changes, such as the abrupt influx of the Laurentide Ice Sheet meltwater to the North Atlantic. The freshwater influx into the N. Atlantic, known to have weakened the Atlantic Meridional Overturning Circulation (AMOC), also led to an enhanced temperature gradient between the two hemispheres, i.e. cold NH and warm SH, shifting the mean position of the ITCZ further south. This brought wet conditions in the SH monsoon regions, such as NW Madagascar, and dry conditions in the NH monsoon regions, including the Asian Monsoon and the East Asian Summer Monsoon.

## 1. Introduction

Although much is known about Holocene climate change worldwide (Mayewski et al., 2004; Wanner and Ritz, 2011; Wanner et al., 2011; 2015), high-resolution climate data for the Holocene period is still regionally limited in the Southern Hemisphere (SH) (e.g., Wanner et al., 2008; Marcott et al., 2013; Wanner et al., 2015). This uneven distribution of data hinders our understanding of the spatio-temporal characteristics of Holocene climate change, and the forcings involved. For example, some of these forcings would have an influence on Inter-Tropical Convergence Zone (ITCZ) behavior and monsoonal response in low- to mid-latitude regions (e.g., Wanner et al., 2015; Talento and Barreiro, 2016). The island of Madagascar, in the southwest Indian Ocean (Fig. 1a), is seasonally visited by the ITCZ with a karst region crossing latitudinal belts (Fig. 1c). Thus, it is a natural laboratory to study changes in the ITCZ over time. New records from Madagascar could fill gaps in paleoclimate reconstruction in the SH that might help refine

paleoclimate simulations, which in turn could provide better understanding of the global
circulation and the land–atmosphere–ocean interaction during the Holocene.
In this paper, we present multiproxy records (stable isotopes, petrography, mineralogy,
variability of layer-specific width, or LSW) from stalagmites from Anjohibe and Anjokipoty caves.
Stalagmites are used because of their potential to store significant climatic information (e.g.,
Fairchild and Baker, 2012, p. 9–10), and in Anjohibe Cave, recent studies have shown the
replicability of paleoclimate records from stalagmites (e.g., Burns et al., 2016).
Two stalagmites were investigated, and these allowed us to characterize Holocene climate
change in NW Madagascar. First, we developed a record of climate change from the multiproxy
data. With a better understanding of Madagascar's paleoclimate, we then investigated possible
climate drivers of tropical climate change to draw a more comprehensive conclusion on the
major factors controlling the hydrological cycle in NW Madagascar and surrounding regions
during the Holocene.

## 76   2. Setting

### 77   *2.1.Stalagmites and their setting*

Stalagmites are secondary cave deposits that are $CaCO_3$ precipitates from cave dripwater.
Calcium carbonate precipitation occurs mainly by $CO_2$ degassing, which increases the pH of the
dripwater and thus increases the concentration of $CO_3^{2-}$. In some cases, evaporation may also
contribute to increased $Ca^{2+}$ and/or $CO_3^{2-}$ concentration in dripwater. $CO_2$ degassing occurs when
high-$P$CO$_2$ water from the epikarst encounters low-$P$CO$_2$ cave air. Evaporation occurs when
humidity inside the cave is relatively low. The fundamental equation for stalagmite deposition is
shown in Eq. 1.
$$Ca^{2+}_{(aq)} + 2HCO^-_{3\,(aq)} \rightleftharpoons CaCO_{3\,(s)} + CO_{2\,(g)} + H_2O_{(l)} \text{ (Eq. 1)}$$
Growth and non-growth of stalagmites depends on conditions that affect the reaction of Eq. 1
above. An increase in $Ca^{2+}$ drives the equation to the right (towards precipitation) and an
increase in $CO_2$ of the cave air and/or $H_2O$ drives it to the left (towards dissolution). All
components of the equation are influenced by the supply of water to the cave, which is generally
climate-dependent. More water enters the cave during warm/rainy seasons than during cold/dry

seasons. Stalagmites will form when cave dripwater is saturated with respect to calcite and/or aragonite. If the water passes through the bedrock too quickly to dissolve significant carbonate rock, and/or enters the cave and reaches the stalagmite too quickly to degas significant $CO_2$, it will not be saturated with respect to $CaCO_3$, inhibiting stalagmite formation. Stalagmite growth will slow as dripwater declines and will stop entirely if flow ceases. Vegetation provides $CO_2$ to the soil via root respiration so the vegetation cover above the cave and the type of vegetation can promote or limit stalagmite growth. Overall, the karst hydrological system plays a crucial role in the deposition and non-deposition of stalagmites, and this is closely linked to changes in local and regional environment and climate.

### 2.2. Regional environmental setting

Stalagmites ANJB-2 and MAJ-5 were collected from Anjohibe and Anjokipoty caves, respectively, in the Majunga region of NW Madagascar (Fig. 1). Sediments and fossils from these caves have already provided many insights about the paleoenvironmental and archaeological history of NW Madagascar (e.g., Burney et al., 1997, 2004; Brook et al., 1999; Gommery et al., 2011; Jungers et al., 2008; Vasey et al., 2013; Burns et al., 2016; Voarintsoa et al., 2017b).

Anjohibe (S15° 32' 33.3"; E046° 53' 07.4") and Anjokipoty (S15° 34' 42.2"; E046° 44' 03.7") are about 16.5 km apart (Fig. 1c). Their location in the zone visited by the ITCZ (e.g., Nassor and Jury, 1998) makes them ideal sites to test the hypothesis that latitudinal migration of the ITCZ influenced the Holocene climate of NW Madagascar (e.g., Chiang and Bitz, 2005; Broccoli et al., 2006; Chiang and Friedman, 2012; Schneider et al., 2014). The ITCZ brings north or northwesterly monsoon winds to Madagascar during austral summers, in a pattern that the Service Météorologique of Madagascar calls the "Malagasy monsoon". Majunga has a tropical savanna climate (Aw) according to the Köppen-Geiger climate classification, with a distinct wet summer (from October to April) and dry winter (May-September). The mean annual rainfall is around 1160 mm. The mean maximum temperature in November, the hottest month in the summer, is about 32°C. The mean minimum temperature in July, the coldest month of the dry winter, is about 18°C (Fig. 1b).

### 2.3. Climate of Madagascar


The climate of Madagascar is unique because of its varied topography and its position in the
Indian Ocean. Some scientists refer Madagascar as a "laboratory" for paleoecological study (e.g.,
Burney, 1997) because it is not only susceptible to several climatic forcing mechanisms but also
an island with recent anthropogenic interaction, living imprints in the geological records (e.g.,
Burney et al., 2003, 2004; Matsumoto and Burney, 1994; Crowley and Samonds, 2013; Burns et
al., 2016; Voarintsoa et al., 2017b). Its climate has been reviewed in several recent works (e.g.,
Jury, 2003; DGM, 2008, Douglas and Zinke, 2015, p. 281-299; Voarintsoa et al., 2017b, p.138-
139; Scroxton et al., 2017).   Regionally distinct rainfall gradients from east to west and from
north to south are evident across the country (Jury, 2003; Dewar and Richard, 2007), and these
are linked to easterly trade-winds in winter (May-October) and northwesterly tropical storms in
summer, respectively. The Malagasy monsoon is modulated by the seasonal north-south
migration of the ITCZ, which is the main driver of austral summer rainfall in Madagascar. The
ITCZ's mean position has shifted northward or southward depending on the global climate
conditions, but most generally it migrates towards the Earth's warmer hemisphere (Frierson and
Hwang, 2012; Kang et al., 2008; McGee et al., 2014; Sachs et al., 2009). A relationship between
this long-term migration of the ITCZ and climate in Madagascar was reported in NW Madagascar
between c. 370 CE and 800 CE (see Fig, 8 of Voarintsoa et al., 2017b).
Beyond ITCZ, climate of Madagascar is also influenced by changes in Indian Ocean sea surface
temperatures (SST) (Zinke et al., 2004; see also Kunhert et al., 2014) and changes in SST of the
adjacent current off southwestern Madagascar, the Aghulas Current (Lutjeharms, 2006; Beal et
al., 2011; Zinke et al., 2014). The most immediate signal is the Indian Ocean Dipole (IOD), or
Indian Ocean Zonal Mode (Li et al., 2003). IOD-like patterns have been proposed as possible
contributors to Holocene climate variability in tropical Indian Ocean (Abram et al., 2009; Tierney
et al., 2013). IOD is as a coupled atmosphere-ocean mode in the tropical Indian Ocean (e.g., Saji
et al., 1999; Webster et al., 1999; Brown et al., 2009; Yagamata et al., 2004; Behera et al., 2013).
It is characterized by a reversal of the climatological SST gradient and winds across the Indian
Ocean basin (Saji et al., 1999; Webster et al., 1999; Abram et al., 2007; Brown et al., 2009). A
positive IOD event starts with anomalous SST cooling along the Sumatra-Java coast in the eastern
Indian Ocean (Abram et al., 2007, 2008), along with positive SST anomaly in the western part of
the basin (e.g., Saji et al., 1999; Abram et al., 2007). Such positive IOD events are observed to
result in increased precipitation, sometimes causing devastating floods, over East Africa (Black et
al., 2003; Saji et al., 1999; Webster et al., 1999; Saji and Yagamata, 2003; Weller and Cai, 2014).
Such events have also enhanced precipitation over the northern part of India, the Bay of Bengal,
Indochina, and southern part of China in 1994 (e.g., Behera et al., 1999; Guan and Yamagata,
2003; Saji and Yagamata, 2003). In the eastern Indian Ocean, a positive IOD is found to intensify
El-Niño related drought, often as severe droughts, over Indonesia (Webster et al., 1999; Weller
and Cai, 2014). It is however, important to note that the relationship between IOD and El-Nino
Southern Oscillation (ENSO) is still debated. While some researchers found no relationships (e.g.,
Saji et al., 1999; Li et al., 2003; Lee et al., 2008), others found some relationships (e.g., Brown et
al., 2009; Schott et al., 2009; Shinoda et al., 2004; Venzke et al., 2000; Abram et al., 2008; Saji
and Yagamata, 2003; Meyers et al., 2007).
Apart from the coral study of Zinke et al. (2004) and the stalagmite study of Scroxton et
al. (2017), very little is known about the effect of the IOD on Madagascar. One objective of this
stalagmite study is to better understand how such mechanisms influenced climate in
Madagascar during the Holocene.

*2.4.* The Holocene in NW Madagascar
Little is hitherto known about Holocene climate change in NW Madagascar nor about the
major drivers of long-term climatic changes there. Most paleoclimate information from this
region covers the last two millennia with more focus on the anthropogenic effects on the
Malagasy ecosystems (e.g., Crowley and Samonds, 2013; Burns et al., 2016; Voarintsoa et al.,
2017b). This is because several studies show that  megafaunal extinctions in Madagascar
coincide with the arrival of humans around 2-3 ka BP (e.g., see Table 1 of Virah-Sawmy et al.,
2010; MacPhee and Burney, 1991; Burney et al., 1997; Crowley, 2010). There are even fewer
long-term paleoclimate records for the NW region, with only sediments from Lake Mitsinjo
(3,500 yr. BP; Matsumoto and Burney, 1994) and stalagmites from Anjohibe Cave (40,000 yr. BP;
Burney et al. 1997) providing records of more than 3 kyr. Even though these records provided
useful information about the paleoenvironmental changes in NW Madagascar, their linkages to
global climatic change, such as the linkages to the ITCZ, are not yet fully understood.
## 3.  Methods
### 3.1. Radiometric dating
A total of 22 samples were drilled from Stalagmite ANJB-2 and 9 samples for Stalagmite
MAJ-5 for U-series dating (Table S1 and S2). Each sample is a long (~5 to 20 mm), narrow (~1-
2mm), and shallow (~1 mm) trench, allowing us to extract 50–250 mg of $CaCO_3$ powder. We
followed the chemical procedures described in Edwards et al. (1987) and Shen et al. (2002) when
separating uranium and thorium. U/Th measurements were performed on the multi-collector
ICP-MS of the University of Minnesota, USA and on a similar instrument in the Stable Isotopes
Laboratory of Xi'an, in Jiaotong, China. Instrument details are provided in Cheng et al. (2013).
Corrected $^{230}$Th ages assume an initial $^{230}$Th/$^{232}$Th atomic ratio of $4.4 \pm 2.2 \times 10^{-6}$. This is the ratio
for "bulk earth" or crustal material at secular equilibrium with a $^{232}$Th/$^{238}$U value of 3.8. The
uncertainty in the "bulk earth" value is assumed to be ±50% (see footnotes to Table S1 and S2).
The error in the final "corrected age" incorporates this uncertainty. The radiometric data are
reported as year BP, where BP is Before Present, and "Present" is A.D. 1950. Stalagmite
chronologies were constructed using the StalAge1.0 algorithm of Scholz and Hoffman (2011) and
Scholz et al. (2012), an algorithm using a Monte-Carlo simulation designed to construct
speleothem age models. The algorithm can identify major and minor outliers and age inversions.
The StalAge scripts were run on the statistics program R version 3.2.2 (2015-08-14). The age
models were adjusted considering hiatal surfaces identified in the samples, using the approach
of Railsback et al. (2013; see their Fig. 9).

### 3.2. Petrography and mineralogy
Petrography and mineralogy of the two stalagmites were investigated 1) by examining
both the polished surfaces and the scanned images of the sectioned stalagmites, and by
identifying any diagenetic fabrics (e.g., Zhang et al., 2014) that could potentially affect stable
isotope values, 2) by observing eleven oversized thin sections (3x2 in) under the Leitz Laborlux
12 Pol microscope and the Leica DMLP equipped with QCapture in the Sedimentary
Geochemistry Lab at the University of Georgia, 3) by  using scanning electron microscopy (SEM)
to better understand the mineralogical fabrics at locations of interest (Fig. S13), and 4) by
analyzing about 30–100 mg of powdered spelean layers (n=15) on a Bruker D8 X-ray
Diffractometer in the Department of Geology, University of Georgia. For calcite and aragonite
identification, we used CoK$\alpha$ radiation at a 2$\theta$ angle between 20° and 60°.

Layer-specific width (LSW) of clearly-defined layers was measured at selected locations

on the stalagmite polished surfaces (Fig. S4; Sletten et al., 2013; Railsback et al., 2014;
Voarintsoa et al., 2017b). LSW is the horizontal distance between two points on the flanks of the
stalagmite where convexity is greatest. It is the width near the top of the stalagmite when the
layer being examined was deposited. LSW is measured at right angles to the growth axis of the
stalagmite; it is the horizontal distance between points on the layer growth surface becomes
tangent to a line inclined at 35° to the growth axis (Fig. S4). LSW may vary along the length of the
stalagmite, with smaller values suggesting drier conditions and larger values wetter conditions.

3.3. Stable isotopes

Stable isotope samples of 50–100 µg were manually drilled along the stalagmite's growth

layers at the crest. The trench size is very small (1.5 x 0.5 x 0.5 mm). Since a small mixture of
calcite and aragonite could potentially change the $\delta^{18}$O and $\delta^{13}$C of the measured spelean layers
(see for example Frisia et al., 2002), drilling and sample extraction was carefully done on
individually discrete layers using the smallest drill-bit head (SSW-HP-1/4) to avoid potential
mixing between calcite and aragonite. The polished surface of the two stalagmites were
examined to see if features of diagenetic alteration are present (see for example fig. 2 of Zhang
et al., 2014), but none was found. During sampling, the mineralogy at the crest, where stable
isotope samples were extracted, was recorded for future mineralogical correction.

Aragonite oxygen and carbon isotopic corrections were performed to compensate for

aragonite's inherent fractionation of heavier isotopes (e.g., Romanek et al., 1992; Kim et al.,
2007; McMillan et al., 2005) and to remove the mineralogical bias in isotopic interpretation
between calcite and aragonite. The correction consists of subtracting 0.8‰ for $\delta^{18}$O (Kim and
O'Neil, 1997; Tarutani et al., 1969; Kim et al., 2007; Zhang et al., 2014) and 1.7 ‰ for $\delta^{13}$C
(Rubinson and Clayton, 1969; Romanek et al., 1992) for the aragonite as has been done
previously (e.g., Holmgren et al., 2003; Sletten et al., 2013; Liang et al., 2015; Railsback et al.,
2016; Voarintsoa et al., 2017a) as shown in equations 2 and 3 below (where $R_{A/C}$ is the aragonite
percentage if not 100%).
$\delta^{18}O_{corr.}$ (‰, VPDB) = $\delta^{18}O_{uncorr.}$ (‰, VPDB) – [$R_{A/C}$ x 0.8 (‰, VPDB)] (Eq. 2)
$\delta^{13}C_{corr.}$ (‰, VPDB) = $\delta^{13}C_{uncorr.}$ (‰, VPDB) – [$R_{A/C}$ x 1.7 (‰, VPDB)] (Eq. 3)
Supplementary Figures S6–S8 show both the corrected and uncorrected isotopic records.
For the analytical methods, oxygen and carbon isotope ratios were measured using the
Finnigan MAT-253 mass spectrometer fitted with the Kiel IV Carbonate Device of the Xi'an Stable
Isotope Laboratory in China (ANJB-2; n=654) and using the Delta V Plus at 50°C fitted with the
GasBench-IRMS machine of the Alabama Stable Isotope Laboratory in USA (MAJ-5; n=286).
Analytical procedures using the MAT 253 are identical to those described in Dykoski et al. (2005),
with isotopic measurement errors of less than 0.1 ‰ for both $\delta^{13}C$ and $\delta^{18}O$. Analytical methods
and procedures using the GasBench-IRMS machine are identical to those described in Skrzypek
and Paul (2006), Paul and Skrzypek (2007), and Lambert and Aharon (2011), with ±0.1 ‰ errors
for both $\delta^{13}C$ and $\delta^{18}O$. In both techniques, the results are reported relative to Vienna PeeDee
Belemnite (VPDB) and with standardization relative to NBS19. An inter-lab comparison of the
isotopic results was conducted, and it involved replicating every tenth sample of Stalagmite MAJ-
5 at both labs. This exercise showed a strong correlation between the lab results (Fig. S5).

## 4. Results

### 4.1. Radiometric data

Results from radiometric analyses of the two stalagmites are presented in Tables S1 and
S2. Corrected [230]Th ages suggest that Stalagmite ANJB-2 was deposited between c. 8977±50 and
c. 161±64 yr. BP, and Stalagmite MAJ-5 was deposited between c. 9796±64 and c. 150±24 yr. BP.
These ages collectively indicate stalagmite deposition at the beginning (between 9.8 and 7.8 ka
BP) and at the end of the Holocene (after c. 1.6 ka BP). In both stalagmites, the older ages have
small 2σ errors and they generally fall in correct stratigraphic order, except sample ANJB-2-120
and its replicate ANJB-2-120R, which were not used because of the sample's high porosity and
high detritals content. In contrast, many of the younger ages have larger uncertainties.  This is
mainly because many of the younger samples have very low uranium concentration and the
detrital thorium concentration is also high, similar to what Dorale et al. (2004) reported. We also
understand that the value for initial $^{230}$Th correction, i.e. the initial $^{230}$Th/$^{232}$Th atomic ratio of 4.4
± 2.2 × 10$^{-6}$ for a bulk earth with a $^{232}$Th/$^{238}$U value of 3.8, in these samples could have slightly
altered the $^{230}$Th age of these younger samples, leading to larger uncertainties (such as discussed
in Lachniet et al., 2012). We encountered similar problems while working on other younger
samples from the same cave, but we compared the stable isotope profile with other published
records using isochron corrections, and results did not differ significantly (see Fig. 9 of
Voarintsoa et al., 2017b). Since this work does not focus on decadal or centennial interpretation
of the Late Holocene stable isotope data, additional chronology adjustment has not been made,
and we used the chronology from StalAge to construct the time series. However, in Figures 5 and
6, age uncertainties are given below the stable isotope profiles so that comparisons with other
records can accommodate these uncertainties.

The key finding from our age and petrographic data for the two stalagmites is that they

suggest that there were three distinct intervals of growth and non-growth during the Holocene
(Figs. 2–4, 7). The information suggesting this includes: (1) CaCO$_3$ deposition between c. 9.8 and
7.8 ka B.P., (2) a long depositional hiatus between c. 7.8 and 1.6 ka B.P., and (3) resumption of
CaCO$_3$ deposition after c. 1.6 ka B.P. In the rest of the paper, we will refer to these intervals as
the Malagasy Early Holocene Interval (MEHI), Malagasy Mid-Holocene Interval (MMHI), and
Malagasy Late Holocene Interval (MLHI), respectively.

*4.2.Stable isotopes*

Raw values of $\delta^{18}$O and $\delta^{13}$C for Stalagmite ANJB-2 range from −8.9 to −2.3‰ (mean = −

5.0‰), and from −11.0 to +5.2‰ (mean = −4.2‰), respectively, relative to VPDB.  Raw values of
$\delta^{18}$O and $\delta^{13}$C for Stalagmite MAJ-5 range from −8.8 to −0.9‰ (mean = −4.9‰), and from −9.4 to
+2.6‰ (mean = −4.4‰), respectively, relative to VPDB. Mean $\delta^{18}$O and $\delta^{13}$C values are
distinguishable between the MEHI and the MLHI. In both stalagmites, the amplitude of $\delta^{18}$O
fluctuations was fairly constant throughout the Holocene; whereas the $\delta^{13}$C profile shows a
dramatic shift toward higher values (i.e. from -10.9‰ to +3.8‰, VPDB) at c. 1.5 ka BP.
The MEHI and MLHI are isotopically distinct (Fig. 4). The MEHI is characterized by statistically
correlated $\delta^{18}$O and $\delta^{13}$C ($r^2$=0.65 and 0.53), and much depleted $\delta^{13}$C values (c −11.0 to −4.0 ‰).
The 8.2 ka event, a widespread cold event in the NH (e.g., Alley et al., 1997), is also apparent in
the stalagmite records. Stalagmite $\delta^{18}$O and $\delta^{13}$C ratios reach their lowest values of −6.8 and −
10.9‰, respectively during that interval (Figs. 5, 12). In contrast to the MEHI, the values of $\delta^{18}$O
and $\delta^{13}$C during the MLHI are poorly correlated ($r^2$=0.25 and 0.17), and $\delta^{13}$C values are more
enriched (Figs. 4, 6).
Since Stalagmites ANJB-2 and MAJ-5 were collected from two different caves 16 km apart,
discrepancies between the stable isotopes at the same age are expected, suggesting that local
conditions could be one of the discrepancy factors. Another potential source for the discrepancy
is the larger uncertainty of the younger ages due to low uranium and high detrital
concentrations. This U-Th aspect has been a challenge for several young stalagmites (e.g., Dorale
et al., 2004; Lachniet et al., 2012) including samples from NW Madagascar (this study). While the
utility of speleothems as a climate proxy largely depends on replication of stable isotope values,
it is important to note that perfect stable isotope replication can only occur between stalagmites
collected from the same cave chamber (e.g., Dong et al., 2010; Burns et al., 2016).

### 4.3. Mineralogy, petrography, and layer-specific width

In both stalagmites, the hiatus of deposition is characterized by a well-developed Type L
surface (Figs. 2, 3, S15). Petrography and mineralogy are distinct before and after this hiatus (Fig.
3). Below the hiatus, laminations are well preserved in both stalagmites. Above the hiatus,
laminations are not well-preserved, although noted in some intervals.
In Stalagmite ANJB-2, the layer-specific width varies from 37 to 26.5 mm with a mean of
30 mm. It decreases to 28 mm at the hiatus (Fig. 3). Below the hiatus, mineralogy is dominated
by aragonite, although a few thick layers of calcite are also identified. A thin (~2-3 mm) but
remarkable layer of white, very soft, and porous aragonite is identified just below the hiatus (Fig.
S15). This layer is covered by a very thin layer of dirty carbonate. Above the hiatus, mineralogy is
also composed of calcite and aragonite, with calcite dominant, and the calcite layers contain
macro-cavities that are mostly off-axis macroholes (Shtober-Zisu et al., 2012).
In Stalagmite MAJ-5, LSW varies from 50 to 22 mm with a mean of 35.5 mm. It decreases
to 22 mm at the hiatus (Fig. 3). Below the hiatus, mineralogy is a mixture of calcite and
aragonite. Above the hiatus, mineralogy is mainly calcite and macro-cavities are also present
throughout that upper part of the stalagmite.

*4.4.*Summary of results
The various records from Stalagmites ANJB-2 and MAJ-5 suggest three distinct
climate/hydrological intervals of the Holocene.  The MEHI (c. 9.8 to 7.8 ka BP), with evidence of
stalagmite deposition, is characterized by statistically correlated $\delta^{18}O$ and $\delta^{13}C$ ($r^2$=0.65 and 0.53)
and more negative $\delta^{13}C$ values (c. −11.0 to −4.0 ‰). The MMHI (c. 7.8 to 1.6 ka BP) is marked by
a long-term hiatus in deposition, which is preceded by a well developed Type L surface in both
Stalagmite ANJB-2 and MAJ-5 (Figs. 3, S15). The Type L surface is observed as an upward
narrowing of the stalagmite's width and layer thickness. It is particularly well developed in
Stalagmite MAJ-5 (Fig. S15). In Stalagmite ANJB-2, the hiatus at the Type L surface is preceded by
a c. 3 mm thick layer of highly porous, very soft, and fibrous white crystals of aragonite (the only
aragonite with such properties). This aragonite is topped by a thin and well-defined layer of
detrital materials (Fig. S15), further supporting the presence of a hiatus. Finally, the MLHI (after
c. 1.6 ka BP) is characterized by poorly correlated $\delta^{18}O$ and $\delta^{13}C$ ($r^2$=0.25–0.17). This interval is
additionally marked by a shift in $\delta^{13}C$ toward higher values (Figs. 4, 6).

5.   Discussion
*5.1.*Paleoclimate significance of stalagmite growth and non-growth: implications for
paleohydrology
Growth and non-growth of stalagmites depends on several factors linked to water
availability, which is largely determined by climate (more water during warm/rainy seasons and
less water during cold/dry seasons). Water is the main dissolution and transporting agent for
most chemicals in speleothems. Cave hydrology varies significantly over time in response to
climate, and this variability influences the formation or dissolution of $CaCO_3$. In this regard,
calcium carbonate does not form if there is little or no water entering the cave, or if there is too
much (see Sect. 2.1). Absence of groundwater recharge most typically occurs during extremely
dry conditions, whereas excessive water input to the cave occurs during extremely wet
conditions. In the latter scenario, water is undersaturated and flow rates are too fast to allow
degassing. Often, water availability is reflected in the extent of vegetation above and around the
cave, as plants require soil moisture or shallow groundwater to survive and propagate, and this
contributes to the stalagmites' processes of formation. The linkage of stalagmites' growth and
non-growth to cave dripwater and soil $CO_2$ is broadly influenced by changes in climate.
Major hiatuses in stalagmite deposition could be marked by a variety of features,
including the presence of erosional surfaces, chalkification, dirt bands/detrital layers, offsetting
of the growth axis, and/or sometimes by color changes (e.g., Holmgren et al., 1995; Dutton et al.,
2009; Railsback et al., 2013; Railsback et al., 2015; Voarintsoa et al., 2017a). Railsback et al.
(2013) were specifically able to identify significant features in stalagmites that allow distinction
between non-deposition during extremely wet (Type E surfaces) and non-deposition during
extremely dry conditions (Type L surfaces; Fig. 3). Physical properties of stalagmites that are
evidence of extreme dry and wet events are summarized in Table 1 of Railsback et al. (2013) and
the mechanism is explained in their Figure 5.
Type E surfaces are layer-bounding surfaces between two spelean layers when the
underlying layers show evidence of truncation. The truncation results from dissolution or erosion
(thus the name "E") of previously-formed layers of stalagmites by abundant undersaturated
water. Type E surfaces are commonly capped with a layer of calcite (Railsback et al., 2013). This
mineralogical trend is not surprising as calcite commonly forms under wetter conditions (e.g.,
Murray, 1954; Pobeguin, 1965; Siegel, 1965; Thrailkill, 1971; Cabrol and Coudray, 1982; Railsback
et al. 1994; Frisia et al., 2002). Additionally, non-carbonate detrital materials are commonly
abundant with varying grain size (i.e., from silt- to sand-size; Railsback et al., 2013).
Type L surfaces, on the other hand, are layer-bounding surfaces where the layers became
narrower upward and thinner towards the flanks of the stalagmite. Decreases in layer thickness
and stalagmites width of the stalagmites upward are indications of lessening deposition (thus the
name "L"; Railsback et al., 2013). Aragonite is a very common mineralogy below a Type L surface,
especially in warmer settings. Layers of aragonite commonly form under drier conditions
(Murray, 1954; Pobeguin, 1965; Siegel, 1965; Thrailkill, 1971; Cabrol and Coudray, 1982;
Railsback et al., 1994; Frisia et al., 2002). Non-carbonate detrital materials are scarce, and if
present, they tend to form a very thin horizon of very fine dust material (Railsback et al., 2013).
Identification of Type L surfaces is aided by measuring the LSW (e.g., Sletten et al., 2013;
Railsback et al., 2014), an approach that is also performed in this study (Fig. S4).

### 5.2. Holocene climate in NW Madagascar

Although the specific boundaries between the Early, Mid, and Late Holocene have been
proposed for global application (Walker et al., 2012; Head and Gibbard, 2015), their use is still
spatially limited (e.g., Wanner et al., 2015). The age models and petrographic features of
Stalagmites ANJB-2 and MAJ-5 suggest three distinct but different Holocene climate intervals
(MEHI, MMHI, and MLHI; see Sect. 4.1) in NW Madagascar. These intervals are illustrated in the
sketches of Figure 4. In this paper, these Malagasy intervals are intended not to argue against
the previously proposed intervals of the Holocene (Walker et al., 2012; Head and Gibbard, 2015).
Instead, they are presented to aid discussion of the available records. For comparison, the
intervals are shown in Fig. 7d.

### 5.2.1.   Malagasy early Holocene interval (c. 9.8 –7.8 ka BP)

Stalagmite deposition during the early Holocene suggests that the chambers, where
stalagmites ANJB-2 and MAJ-5 were collected, were sufficiently supplied with water to allow
$CaCO_3$ precipitation, in accord with Eq.1. This in turn implies relatively wet conditions that could
indicate longer summer rainy seasons relative to modern climate, or wet years in NW
Madagascar (see Supplementary Text 4 and Fig. 8). The correlative $\delta^{13}C$ and $\delta^{18}O$ values further
suggest that vegetation consistently responded to changes in moisture availability, which in turn
was dependent on climate.
One striking aspect of the Stalagmite ANJB-2 $\delta^{18}O$ and $\delta^{13}C$ records is that they parallel
the $\delta^{18}O$ of the Greenland ice core records at c 8.2 ka BP (Figs. 5 and 12). An X-ray diffraction

spectrum for this period, at 195–202 mm from the top of the stalagmite, suggests that the mineralogy at 8.2 ka BP is 100% calcite (Figs. S14, S16–S17). This calcite is not a diagenetic product of aragonite for three reasons. First, the laminations in the thick layer of calcite were not altered (Figs. S16–S17). Second, the polished surface of the stalagmite shows no evidence of fiber relicts and textural ghosts such as observed in Juxtlahuaca Cave in southwestern Mexico (Lachniet et al., 2012) and in Shennong Cave in southeastern China (Zhang et al., 2014). Third, petrographic comparison with known examples of primary and secondary calcite observation under microscope (e.g., Railsback, 2000; Perrin et al., 2014) suggests that there is no strong evidence of aragonite–to–calcite transformation. The decrease in $\delta^{18}$O and $\delta^{13}$C values and the presence of calcite mineralogy at the same interval combine to suggest a wet 8.2 ka BP event in NW Madagascar. The 8.2 ka BP event is a prominent cold event in the North Atlantic records and many NH terrestrial records. It may have been triggered by a release of freshwater from the melting Laurentide Ice Sheet into the North Atlantic basin (e.g., Alley et al., 1997; Barber et al., 1999). Freshwater influx to the Atlantic could have altered the Atlantic Meridional Overturning Circulation (AMOC, e.g., Clark et al., 2001), and could eventually have influenced the climate of Madagascar (Sect. 5.5). Our records reveal a strong link between paleoenvironmental changes in Madagascar and abrupt climatic events in the NH records, suggesting causal relationships.

The MEHI terminated when conditions became much drier, as suggested by increasing $\delta^{18}$O and $\delta^{13}$C values in Stalagmite ANJB-2, by decreasing LSW of both stalagmites, and by major Type L surfaces in both stalagmites. The thin (c. 3 mm), porous, and white aragonite layer in Stalagmite ANJB-2, a very similar deposit to that described in Niggemann et al. (2003), suggests that the terminal drought was at times severe. Aragonite is a $CaCO_3$ polymorph that forms preferentially under drier conditions (Murray, 1954; Pobeguin, 1965; Siegel, 1965; Thrailkill, 1971; Cabrol and Coudray, 1982; Railsback et al. 1994; Frisia et al., 2002). The porous aragonite layer in Stalagmite ANJB-2 is capped by a very thin layer of non-carbonate, brown detritus, which may have been transported to the stalagmite as an aerosol and accumulated on the dry stalagmite surface over time. Accumulation of the detritus must take place in the absence of dripwater (e.g., Railsback et al., 2013). A shift to drier conditions is also supported by isotopic data from Stalagmite ANJ94-5 from Anjohibe Cave (Wang and Brook, 2013; Wang, 2016) in

which relatively low $\delta^{13}$C and $\delta^{18}$O values prior to 7600 BP give way to episodically greater values
thereafter.

*5.2.2.* Malagasy mid-Holocene interval (c. 7.8–1.6 ka BP)
The only data we have for the MMHI is the long term (~6.5 ka) depositional hiatus in both
stalagmites (Figs. 2–3), that potentially indicate dry conditions. The question is why did neither
stalagmite grow during the MMHI? Here, we try to explain the factors and the climatic
conditions that may have been responsible for it.
The documented severe dry conditions at the end of the MEHI (see Sect. 5.2.1) could
have had a significant influence (1) on the cave hydrological system (e.g., Fig. 5 of Asrat et al.,
2007; Bosak, 2010), such as the water conduits (primary or secondary porosity) to the chambers,
and (2) on the vegetation cover above the caves, particularly above the chambers where
Stalagmites ANJB-2 and MAJ-5 were collected. On one hand, it is possible that the dry conditions
late in the MEHI could not only bring lesser water recharge to the cave, but also lowered the
hydraulic head, and increased the rate of evapo-transpiration in the vadose zone. This condition
possibly allowed more air to penetrate the aquifer, perhaps enhancing prior carbonate
precipitation (PCP) in pores and conduits above the caves (e.g., Fairchild and McMillan, 2007;
Fairchild et al., 2000; Johnson et al., 2006; Karmann et al., 2007; McDonald et al., 2007). This
process must have blocked water moving towards Stalagmites ANJB-2 and MAJ-5.  On the other
hand, the late MEHI drying trend (Sect. 5.2.1) could have challenged vegetation to grow, and we
assume that some areas above Anjohibe and Anjokipoty caves must have been devoid of
vegetation. Consequently, biomass activities could have been reduced. Because vegetation
contributes $CO_2$ to the carbonic acid dissolving $CaCO_3$, its absence in certain areas above the
cave could decrease the pH of the percolating water, and perhaps dissolution did not occur.
Under these conditions, even if water reached the stalagmites, it may not have precipitated
carbonate.
Whatever factors were responsible for the long term-depositional hiatus in Stalagmite
ANJB-2 and MAJ-5, we believe that the hiatus was caused by disturbances to water catchments
that feed the chambers at Anjohibe and Anjokipoty caves. The disturbances could be inherited
from the very dry conditions at the end of the MEHI, and/or due to the lack of water supply,
perhaps associated with an increase in epikarst ventilation, and/or by the absence of vegetation.
Water and vegetation are two components of the karst system that play an important role in
$CaCO_3$ dissolution and precipitation (see Eq. 1). Their disturbance may have limited limestone
dissolution in the epikarst and then carbonate precipitation in the cave zone.
Other evidence supports the idea of at least episodic dryness during the MMHI. A work
on a 2-meter long stalagmite (ANJ94-5) from Anjohibe Cave suggests episodic dryness during the
MMHI and a depositional hiatus around the time when Stalagmites ANJB-2 and MAJ-5 stopped
growing (Wang and Brook, 2013; Wang, 2016). For regional comparison, dry spells were also felt
in Central and Southeastern Madagascar (e.g., Gasse and Van Campo, 1998; Virah-Sawmy et al.,

2009).

In summary, several lines of evidence suggest relatively drier climate in NW Madagascar
during the MMHI compared to the MEHI. Drier intervals generally imply drier summer seasons
with less rainfall (Fig. 8), perhaps reflecting shorter visits by the ITCZ. In this regard, even though
the region received rainfall, the necessary conditions could not have been attained to activate
the growth of Stalagmites ANJB-2 and MAJ-5, thus the hiatuses.

*5.2.3.* Malagasy Late Holocene Interval (c. 1.6 ka–present)
Resumption of stalagmite deposition after c. 1.6 ka BP suggests a wetter climate in NW
Madagascar with reactivation of the previous epikarst hydrologic system. Conditions must have
been similar to those of the early Holocene. Wet conditions between c. 850 and 1100 AD in
Stalagmite ANJB-2 and Stalagmite MAJ-5, specifically coincide with glacial advances at northern
high latitudes (Holzhauser et al., 2005) and a cooler interval of the Medieval Climate Anomaly, as
suggested by a negative temperature Anomaly in the NH (e.g., Büntgen et al., 2011; Mann et al.,
1998; Mann and Bradley, 1999, see also Fig. S18).  The sudden beginning of stalagmite growth
during the MLHI and the large $\delta^{13}C$ shift from depleted to enriched values at c. 1.5 ka BP (Fig. 6),
after such long hiatuses may have been associated with changes in vegetation cover above the
cave linked to recent human activities (e.g., Burns et al., 2016; Crowley and Samonds, 2013;
Crowther et al., 2016; Voarintsoa et al., 2017b). Lower $\delta^{13}C$ values in Stalagmite MAJ-5 after 0.8
ka BP (Fig. 3), compared to higher values in Stalagmite ANJB-2, suggests different conditions in or
above the two caves. More human disturbance at one site could account for the different
trends, or alternatively changes in cave micro-climate, or in the hydrologic catchments of the
two stalagmites.

Although the stalagmite data indicate overall wetter conditions during the last c. 1.6 kyr,

there were occasional dry periods, as suggested by several positive peaks in the stalagmite $\delta^{18}O$
records. Drier intervals during the Late Holocene are observed in the Anjohibe data between c.
AD 755 and 795 (i.e., 1195–1155 yr. BP; Voarintsoa et al., 2017b). Similar conditions have been
recorded in other paleoenvironmental studies, in which a peak drought c. 1300–950 cal BP was
reported (Burney, 1987a, b; Burney, 1993; Matsumoto and Burney, 1994; Virah-Sawmy et al.,

2009).


*5.3.*Holocene climate in NW Madagascar: implications for ITCZ dynamics

Figures 7 and 8 depict possible conditions in NW Madagascar during the MEHI, the

MMHI, and the MLHI. Figure 9 summarizes the possible forcings mechanisms linked to the
latitudinal migration of the ITCZ.

In NW Madagascar, stalagmite deposition during the MEHI and the MLHI could suggest

there was sufficient dripwater for stalagmite growth and therefore wetter conditions. This could
have been linked to a more southerly mean position of the ITCZ. Factors that could influence the
mean position of the ITCZ include changes in insolation (e.g., Haug et al., 2001; Wang et al.,
2005; Cruz et al., 2005; Fleitmann et al., 2003, 2007; Schefuß et al., 2005; Suziki, 2011; Kutzbach
and Liu, 1997; Partridge et al., 1997; Verschuren et al., 2009; Voarintsoa et al., 2017a) and
difference in temperature between the two hemispheres (e.g., Chiang and Bitz, 2005; Broccoli et
al., 2006; Chiang and Friedman, 2012; Kang et al., 2008; McGee et al., 2014; Talento and
Barreiro, 2016).

In contrast, the depositional hiatuses during the MMHI could suggest drier conditions,

and thus a northward migration of the mean ITCZ. It seems to agree with the paleoclimate
simulation of Braconnot et al. (2007) of the 6 ka event, suggesting that the NH insolation
increased (Braconnot et al., 2000; see also Chiang, 2009). This northward shift in the mean
position of the ITCZ is consistent with drier conditions, i.e. weaker South American Summer
Monsoon (e.g., Cruz et al., 2005; Seltzer et al., 2000; Wang et al., 2007; but see also Fig.  9 of
Zhang et al., 2013) but wetter conditions in the northern tropics (e.g., Dykoski et al., 2005;
Fleitmann et al., 2007; Gasse, 2000; Haug et al., 2001; Weldeab et al., 2007; Zhang et al., 2013).

5.4. Regional comparisons
Despite differences in Holocene paleoclimate reconstructions for southern Africa,
comparison of the NW Madagascar records with records from neighboring locations (Figs. 10–
11; Table S3) shows that the Holocene wet/dry/wet succession reported in this study has also
been identified at other locations. For example, hydrogen isotope compositions of the n-C31
alkane in GeoB9307-3 from a 6.51 m long marine sediment core retrieved about 100 km off the
Zambezi delta suggest a similar wet/dry/wet climate during Early, Middle, and Late Holocene
respectively (Schefuß et al., 2011). Those changes correspond to changes in temperature from
~26.5° to 27.25° to 27°C, respectively, in the Mozambique Channel, as suggested by alkenone
SST records from sediment cores MD79257 (Bard et al., 1997; Sonzogni et al., 1998). The
Zambezi catchment is specifically relevant here because it is located at the southern boundary of
the modern ITCZ, and so has similar climatic setting as NW Madagascar, and its sensitivity to the
latitudinal migration of the ITCZ could parallel that of Madagascar. Likewise, temperature
reconstruction from the Mozambique Channel could be used to link regional changes in
paleorainfall with regional changes in temperature. A general overview of the Holocene climate
in the African neighboring locations to Madagascar suggests a roughly consistent wetter and
drier climate during the early and middle Holocene, respectively (Fig. 11, Table S3, also see
Gasse, 2000; Singarayer and Burrough, 2015). However, Late Holocene paleoclimate
reconstructions vary. A single answer to this variability is unlikely, but several overlapping
factors, including the latitudinal migration of the ITCZ, changes in ocean oscillations and sea
surface temperatures, volcanic aerosols, and anthropogenic influences could have played a
major role in such variability (e.g., Nicholson, 1996; Gasse, 2000; Tierney et al., 2008; Truc et al.,
2013). Assessing these factors is beyond the scope of this study.

## 5.5. The 8.2 ka event in Madagascar: linkage to ITCZ and AMOC

The 8.2 ka event was a significant short-lived cooling of the N Atlantic and NH during the

Early Holocene (Alley et al., 1997). It is apparent in the ANJB-2 and MAJ-5 stalagmite records as a
wet interval (Sect. 5.2.1; Figs. 5, 12). The 8.2 ka event is a known interval of abrupt freshwater
influx from the melting Laurentide Ice Sheet into the North Atlantic (Alley et al., 1997; Barber et
al., 1999; Kleiven et al., 2008; Carlson et al., 2008; Renssen et al, 2010; Wiersma et al., 2011;
Wanner et al., 2015). It is equivalent to the sharp peak of Bond cycle number 5 (Bond et al. 1997,
2001). This influx of meltwater altered the density and salinity of the NADW. Thornalley et al.
(2009) report that there was a decrease in NADW salinity to approximately 34 p.s.u. during the
Early Holocene.

Understanding the AMOC's influence on Madagascar's hydroclimate could help us better

understand global atmospheric and oceanic circulation, particularly in the SH. An increase in the
flow of freshwater to the North Atlantic decreases the formation of North Atlantic Deep Water,
reducing the meridional heat transport (Barber et al., 1999; Clark et al., 2001; Daley et al., 2011;
Vellinga and Wood 2002; Dong and Sutton 2002, 2007; Dahl et al. 2005; Zhang and Delworth
2005; Daley et al., 2011; Renssen et al., 2001). Weakening of the AMOC would ultimately cause a
widespread cooling in the NH regions (e.g., Clark et al., 2001; Thomas et al., 2007) but warming
in the SH regions (Wiersma et al., 2011; Wiersma and Renssen, 2006). This "cold NH–warm SH"
climate response is similar to the "bipolar seesaw" effect, well-known during the last glacial (e.g.,
Crowley, 1992; Broecker, 1998). The interhemispheric temperature difference between the NH
and SH from such effect could be the driver of the southward displacement of the mean position
of the ITCZ during the 8.2 ka abrupt cooling event. This in turn could have led to an intensified
Malagasy monsoon in NW Madagascar during austral summers, a phenomenon identical to the
South American Summer Monsoon identified in Brazil (e.g., Cheng et al., 2009). In contrast,
regions in the NH monsoon regions became dry at 8.2 ka BP as the Asian Monsoon and the East
Asian Monsoon became weaker (e.g., Wang et al., 2005; Dykoski et al., 2005; Cheng et al., 2009;
Liu et al., 2013).

## 5.6. Beyond the ITCZ: IOD and ENSO influence on Madagascar's climate

Although the ITCZ is the main driver of rainfall availability in Madagascar, recent studies have

also suggested the importance of SST changes in the surrounding ocean and teleconnection with

other climatic phenomena. Scroxton et al. (2017) linked rainfall changes in eastern Indian Ocean

with expansion and contraction of the ITCZ along with positive IOD. Zinke et al. (2004) revealed

strong Indian Ocean subtropical dipole events that were in phase with ENSO indices between AD

1880 and 1920, and between 1930 and 1940, and after 1970 in austral summers. Brook et al.

(1999, p. 700) suggested linkages between rainfall and ENSO in NW Madagascar since AD 1550, a

relationship that is less clear and complicated. This complication could be associated with an

unclear or yet a limited understanding of the relationship between IOD and ENSO, which is not

yet fully understood (e.g., Saji et al., 1999; Li et al., 2003; Lee et al., 2008 versus Brown et al.,

2009; Schott et al., 2009; Shinoda et al., 2004; Venzke et al., 2000; Abram et al., 2008; Saji and

Yagamata, 2003; Meyers et al., 2007).

Our understanding of the oceanic and atmospheric circulation is challenged because IOD and

ENSO share similar features in the associated SST and precipitation anomalies (e.g., Saji et al.,

1999; Webster et al., 1999; Krishnamurty and Kirtman, 2003; Meyers et al., 2007). In addition,

the driving mechanisms of ENSO and IOD during the Holocene are not fully understood, even

though linkages with insolation were reported (e.g., Otto-Bliesner et al., 2003; Liu et al., 2000;

Timmermann et al., 2007; Zheng et al., 2008; Tudhope et al., 2001; Moy et al., 2002; Koutavas et

al., 2006; Conroy et al., 2008; Kuhnert et al., 2014; Liu et al., 2003; Abram et al., 2007). The IOD

signals in the tropical Indian Ocean may additionally be overridden by the global mean

temperature (e.g., Vecchi and Soden, 2007; Zheng et al., 2013), or the signals could be strongly

influenced by monsoonal changes in the surrounding landmasses (e.g., Abram et al., 2007; Qiu et

al., 2012).

Despite the complicated relationships, it is possible that climate of NW Madagascar has been

influenced by ITCZ, IOD, and ENSO, but this is still poorly understood during the Holocene. We

are aware that the temporal and spatial resolution of available records make this investigation

challenging, and we understand that the range of uncertainty of radiometric ages of several

paleoclimate data could be another barrier to fully evaluate such relationship (see for example

Fig. 7 of Kuhnert et al., 2014).

## 6. Conclusions

Petrography, mineralogy, and stable isotope records from Stalagmite ANJB-2, from Anjohibe Cave, and Stalagmite MAJ-5, from Anjokipoty Cave, combine to suggest three distinct intervals of changing climate in Madagascar during the Holocene: relatively wet conditions during the MEHI, relatively drier conditions, possibly due to episodic dryness, during the MMHI, and relatively wet conditions during the MLHI. The timing of stalagmite deposition during the MEHI and the MLHI in NW Madagascar could be attributed to a more southward migration and/or an expanded ITCZ, increasing the duration of the summer rainy seasons, perhaps linked to a stronger Malagasy monsoon. This could have been tied to insolation, the temperature gradient between the two hemispheres, and weakening of the AMOC. In contrast, the c. 6500 year depositional hiatus during the MMHI could indicate a northward migration of the ITCZ, leading to relatively drier conditions in NW Madagascar. The evidence of the 8.2 ka event in the Malagasy records further suggests a strong link between paleoenvironmental changes in Madagascar and abrupt climatic events in the NH, suggesting that during the MEHI Madagascar's climate was very sensitive to abrupt ocean-atmosphere events in the NH.

Although the ITCZ is one of the climatic drivers influencing climate in Madagascar and its surrounding locations, several climatic factors need to be investigated in more detail. For example, we do not fully understand if the latitudinal migration is paired with the expansion and/or expansion of the ITCZ, responsible to changes in several monsoon systems. In addition, the interplay between ITCZ and other factors involving changes in sea surface temperatures, particularly IOD-ENSO, needs to be investigated in details. Data-model comparison seems to be an approach to better understand such relationship. The lack of spatial and temporal resolution of paleoclimate records is still a challenge to fully understand the climate system during the Holocene.

### Author Contribution

N.R.G.V. conceived the research and experiments. N.R.G.V, G.K, A.F.M.R, and M.O.M.R did the fieldwork and collected the samples. X.L., G.K., H.C., R.L.E, and N.R.G.V contributed to the $^{230}$Th dating analyses. N.R.G.V provided detailed investigation of the two stalagmites, provided

stable isotope measurements, prepared thin sections, and conducted X-ray diffraction analyses. G.K. also assisted with the isotopic measurements on Stalagmite ANJB-2. N.R.G.V. wrote the first draft of the manuscript and led the writing. L.B.R. and G.A.B. provided a thorough review of the draft. N.R.G.V. and L.B.R. discussed and revised the manuscript, with additional comments from L.W. N.R.G.V revised the paper with input from all authors, reviewers, and editors.

## Competing Interests

The authors declare no conflict of interest.

## Acknowledgments

This work was supported by grants from (1) the National Natural Science Foundation of China (NSFC 41230524, NBRP 2013CB955902, and NSFC 41472140) to Hai Cheng and Gayatri Kathayat, (2) the Geological Society of America Research Grant (GSA 11166-16) and John Montagne Fund Award to N. Voarintsoa, (3) the Miriam Watts-Wheeler Graduate Student Grant from the Department of Geology at UGA to N. Voarintsoa, and (4) the International Association of Sedimentology Post-Graduate Grant to N. Voarintsoa. We also thank the Schlumberger Foundation for providing additional support to N. Voarintsoa's research. We thank the Department of Geology at the University of Antananarivo, in Madagascar, the Ministry of Energy and Mines, the local village and guides in Majunga for easing our research in Madagascar. We specifically thank Prof. Voahangy Ratrimo, former Department Head of the Department of Geology at the University of Antananarivo, for collaborating with us and for giving us permission to conduct field expedition in Madagascar. We thank Prof. Paul Schroeder for giving us access to use the X-ray diffractometer of the Geology Department to conduct analysis on the mineralogy of the two stalagmites. We thank Prof. John Shields of the Georgia Electron Microscope, University of Georgia, for giving Voarintsoa access to use the Zeiss 1450EP (Carl Zeiss, Inc., Thornwood, NY) for SEM purposes. We also thank Prof. Sally Walker for allowing us to use the microscope of the paleontology lab and for helping us photograph the stalagmites at very high resolution. We also thank Prof. John Chiang of the University of California at Berkeley, for sharing his thoughts and guiding us to literature of relevance to this study.

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

Figures

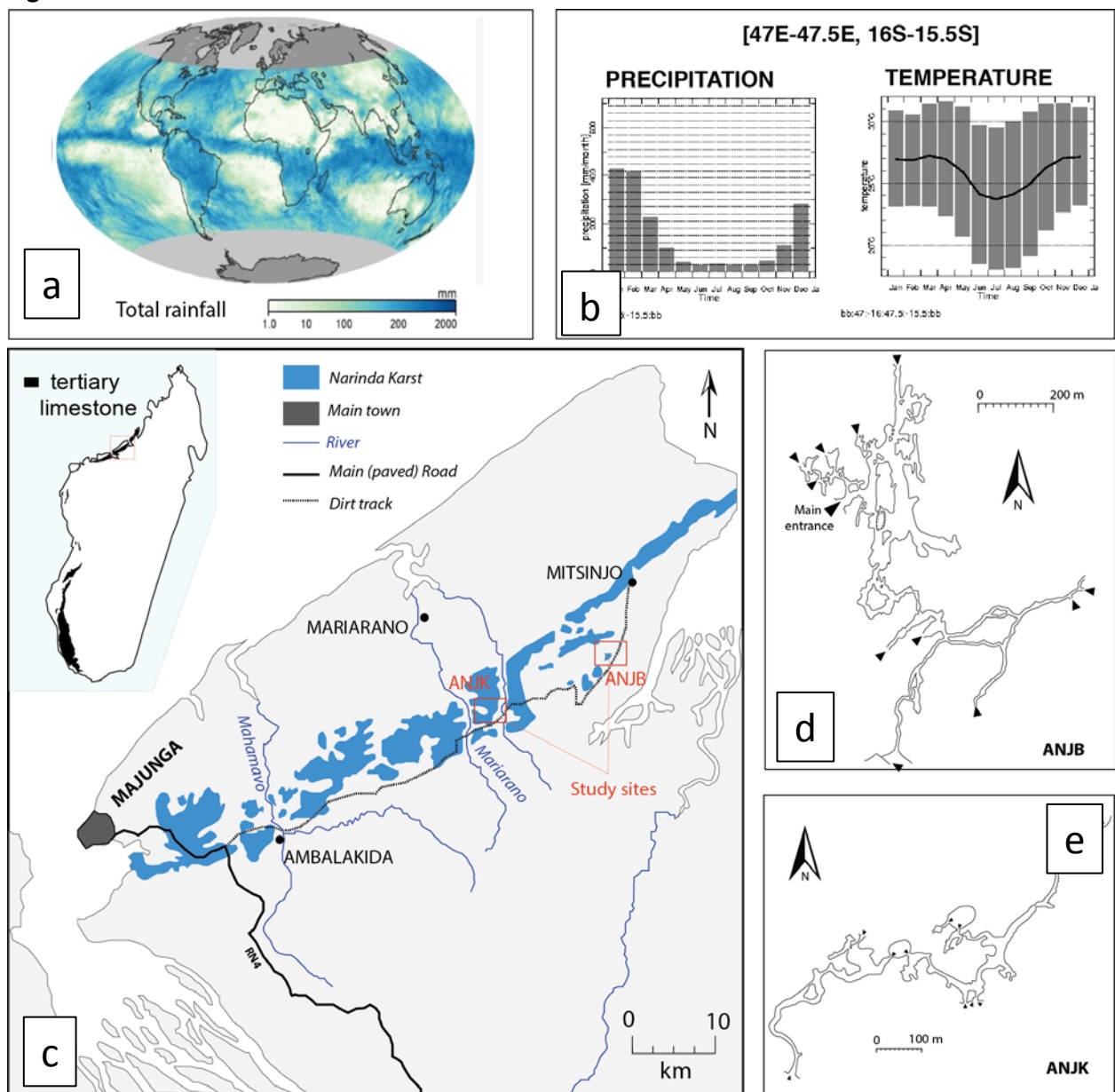


Figure 1: **Climatological and geographic setting of Madagascar and the study area**. (a) Global
rainfall maps recorded by NASA's Tropical Rainfall Measuring Mission (TRMM) satellite showing
the total monthly rainfall in millimeters and the overall position of the ITCZ during November,
2006. Darker blue shades indicate regions of higher rainfall (source: NASA Earth Observatory,
2016). (b) Barplots of the monthly climatology of precipitation, and the monthly average of daily
maximum, minimum, and mean temperature in NW Madagascar. The base period used for the
climatology is 1971-2000. Source: http://iridl.ldeo.columbia.edu/ (accessed August 31, 2016). (c)
Simplified map showing the southwest part of the Narinda karst and the location of the study
areas. Inset figure is a map of Madagascar showing the extent of the Tertiary limestone cover
that makes up the Narinda karst. (d-e) Maps of Anjohibe (ANJB) and Anjokipoty (ANJK) caves (St-
Ours, 1959; Middleton and Middleton, 2002). See Figs. S1–S3 for additional information about
the study locations.

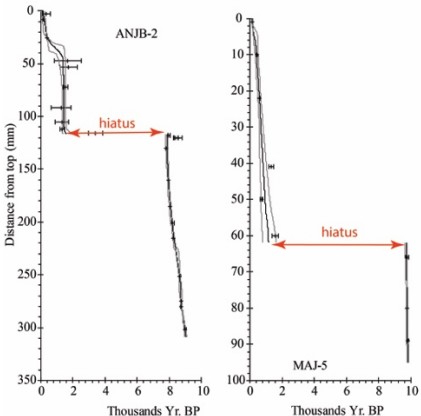


Figure 2: **Age model of Stalagmite ANJB-2 and MAJ-5** using the StalAge1.0 algorithm of Scholz and
Hoffman (2011) and Scholz et al. (2012).


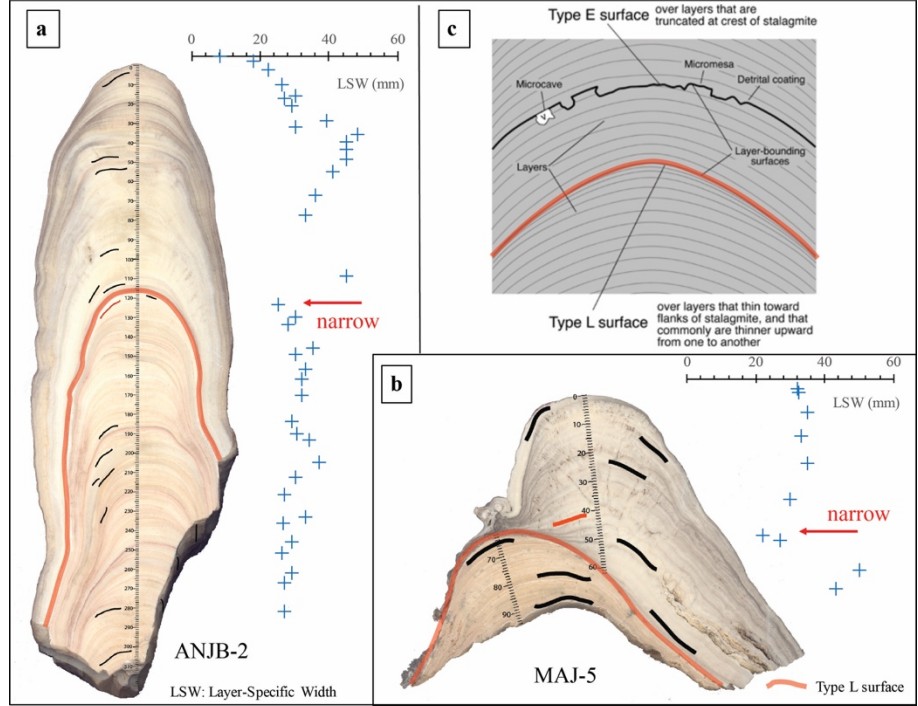



Figure 3: a) Scanned image of Stalagmite ANJB-2 and the corresponding variations in layer-
specific width (LSW). b) Scanned image of Stalagmite MAJ-5 and the corresponding layer-specific
width (LSW). c) Sketches of typical layer-bounding surfaces (Type E and Type L) of Railsback et al.
(2013). Close-up of photographs of the hiatuses are shown in Fig S6.

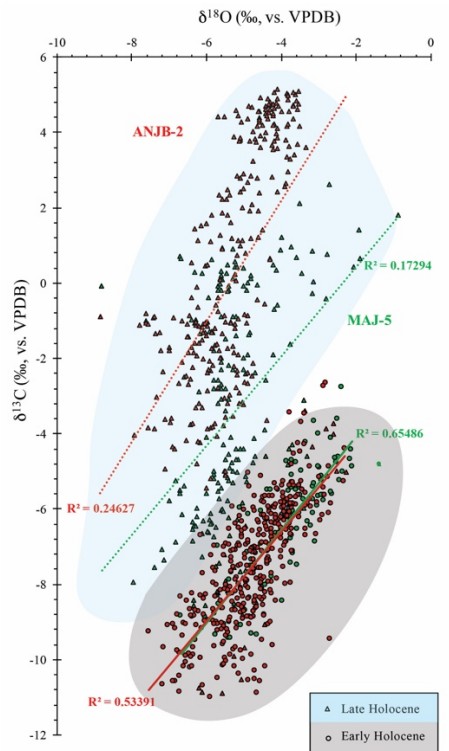


Figure 4: **Stable isotope data**. Scatterplots of $\delta^{13}$C and $\delta^{18}$O for Stalagmite MAJ-5 (green) and
ANJB-2 (red) during the Malagasy early Holocene interval (circle) and the Malagasy late Holocene
interval (triangle). The plot shows distinctive early and late Holocene conditions (roughly
highlighted in gray and light blue shade, respectively).




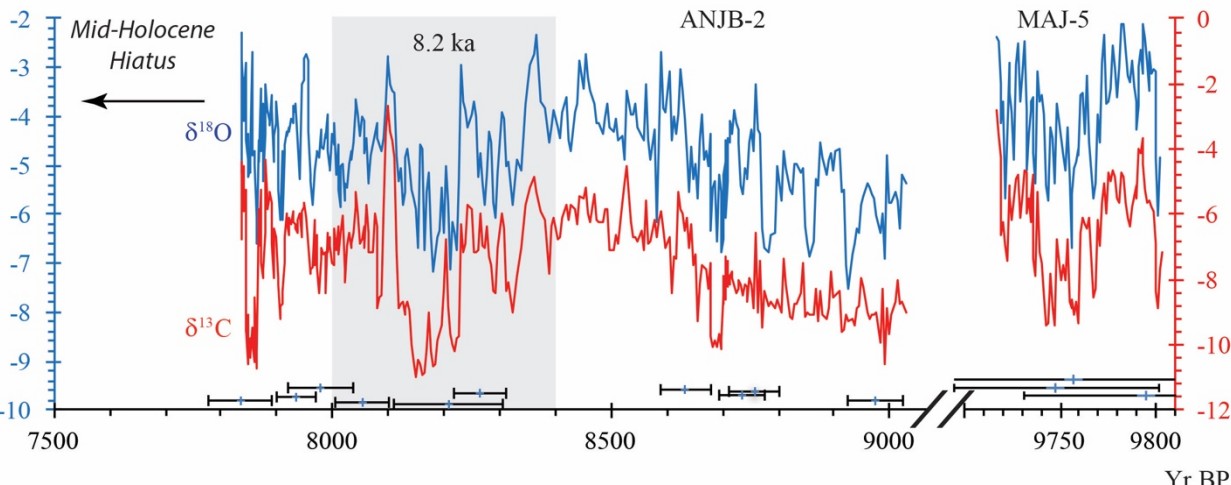


Figure 5: Variations in $\delta^{13}C$ and $\delta^{18}O$ in Stalagmite ANJB-2 and Stalagmite MAJ-5 during the
Malagasy Early Holocene Interval. Supplementary Fig. S6 shows both the corrected and
uncorrected values.

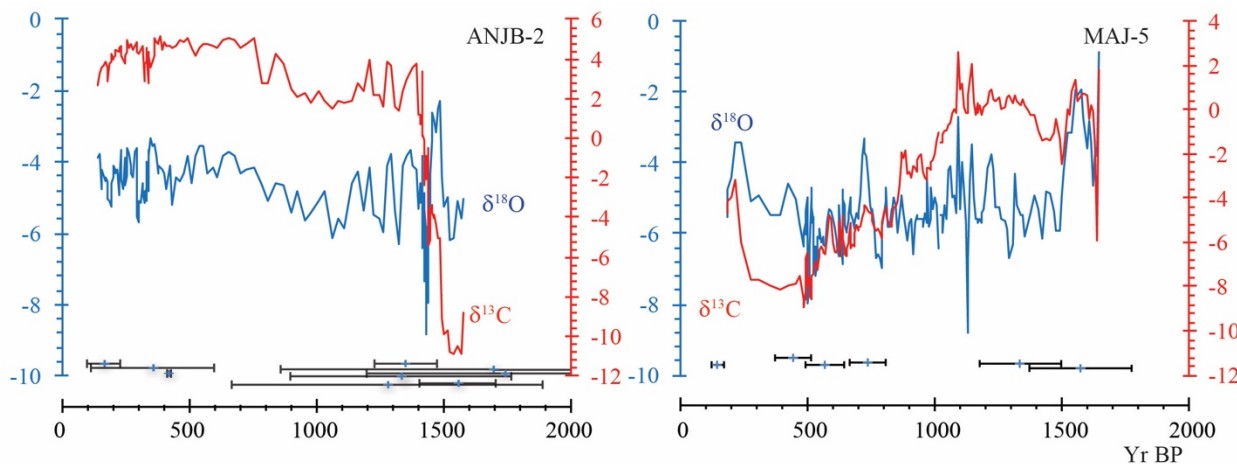


Figure 6: Variations in $\delta^{13}C$ and $\delta^{18}O$ in Stalagmite ANJB-2 and Stalagmite MAJ-5 during the
Malagasy Late Holocene Interval. Supplementary Fig. S7 shows both the corrected and
uncorrected values, and Fig. S8 compares the corrected $\delta^{18}O$ for both stalagmites.



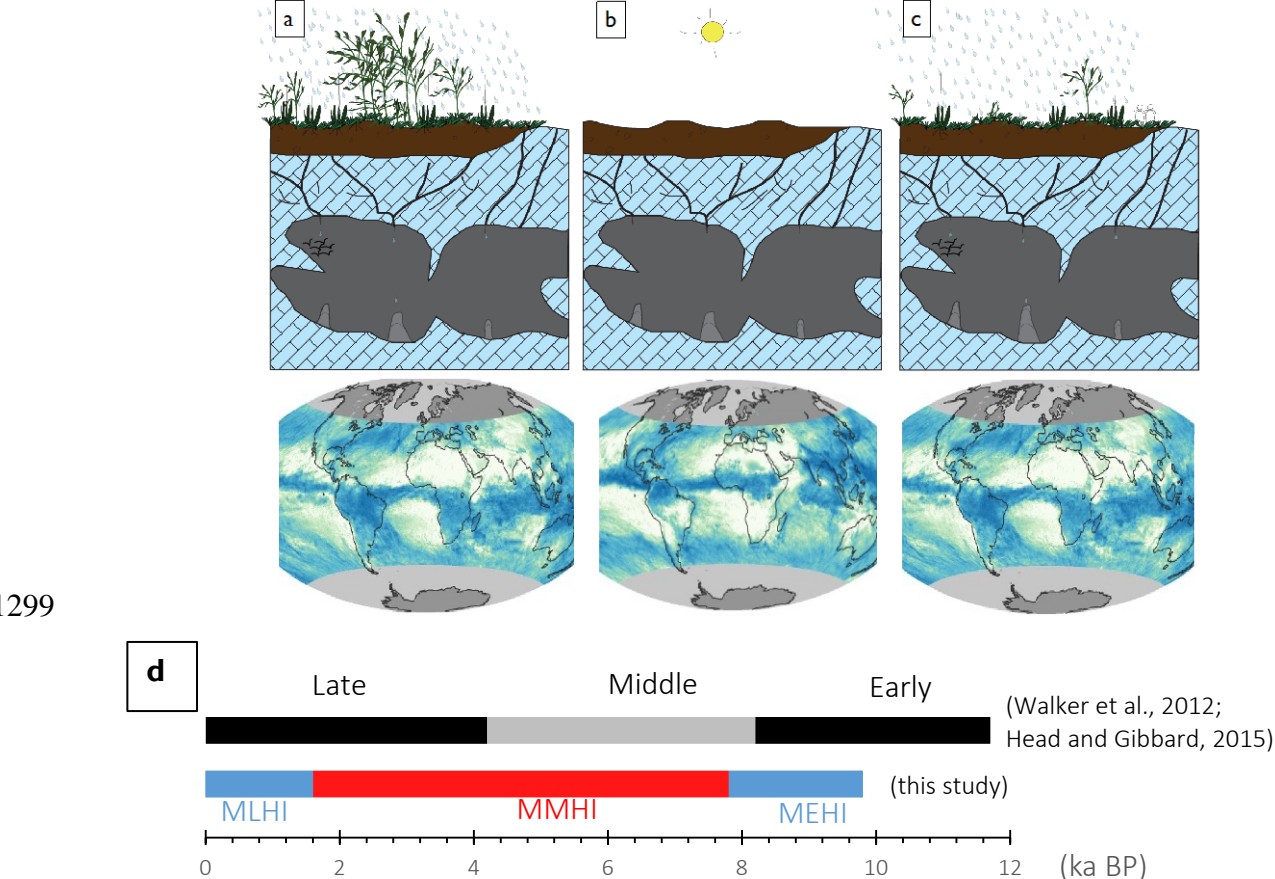


Figure 7: Simplified models portraying the Holocene climate change in NW Madagascar and the possible climatic conditions linked to the ITCZ. a) Wetter conditions during the early Holocene with ITCZ south (prior to c 7.8 ka), favorable for stalagmite deposition. b) Periodic dry conditions during the mid-Holocene (between c. 7.8 and 1.6 ka) with ITCZ north with no stalagmite formation (refer to Sect. 5.2.2). c) Wetter conditions during the late Holocene (after c. 1.6 ka) with ITCZ south, favorable for stalagmite deposition. For details about paleo-vegetation reconstruction. Drawings are not to scale. The bottom figures are from the same source as Fig. 1a, and they are only used here to give a perspective of the possible position of the ITCZ during the early, mid, and late Holocene. d) Comparison of the three Malagasy Holocene interval with the Walker et al. (2012) and Head and Gibbard (2015) subdivision (see text for details, Sect. 5.2).

1311

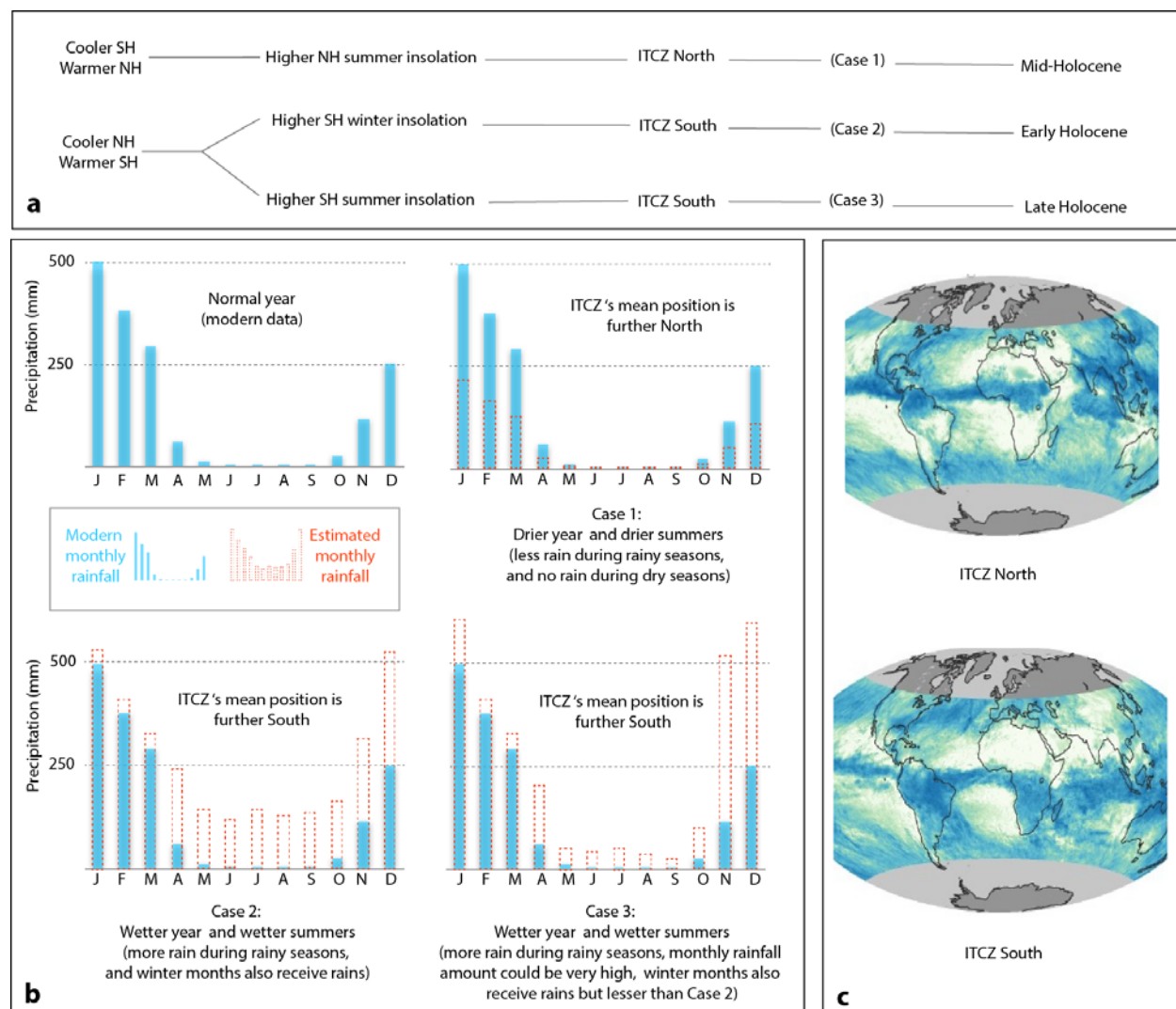

Figure 8: **Conceptualizing the different possible outcomes of the long-term latitudinal migration of the ITCZ**. a) Highlighting the three possible scenarios of the Holocene. b) Barplots of monthly rainfall in NW Madagascar, using the modern data as a reference to estimating the region's paleoclimate during drier and wetter conditions. c) Global rainfall maps from NASA (same source as Fig.1). These maps are modern, but they are only shown here to give a better perspective of the position of Madagascar when the ITCZ is relatively north or south. See supplementary text for details.

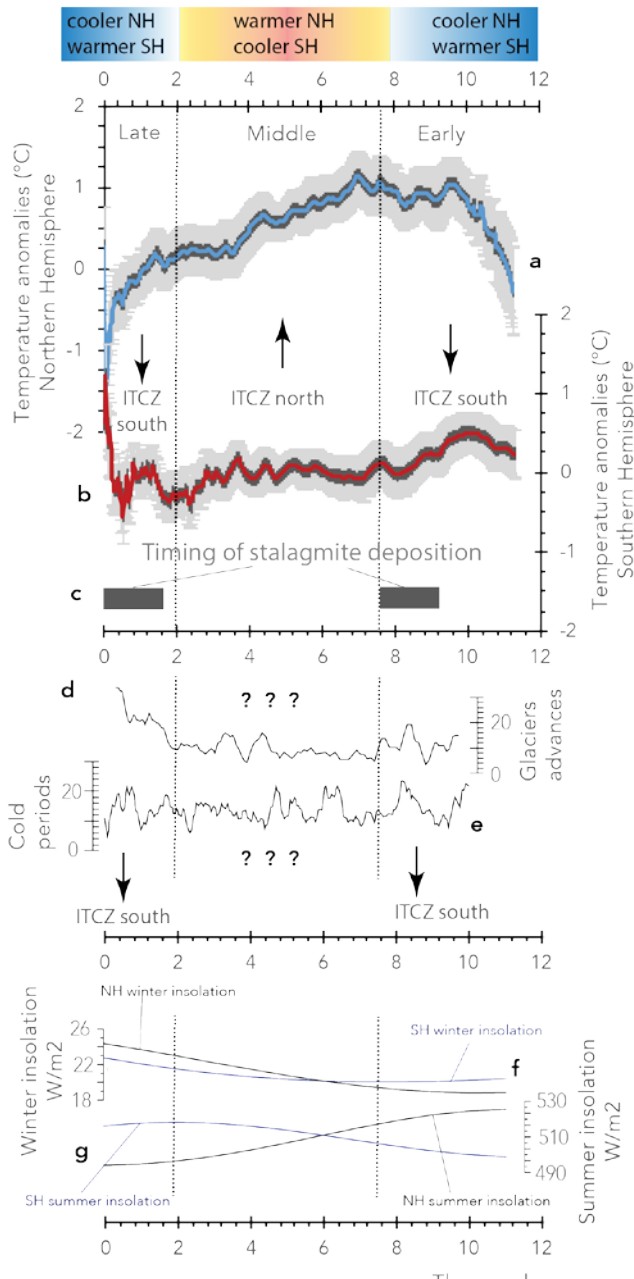

Figure 9: **Possible Holocene climate forcings that influenced climate of NW Madagascar**. a) Average Holocene temperatures in the NH 90°–30°N (blue). b) Average Holocene temperatures in the SH 90°–30°S (red). These temperature data are referenced to the 1961–1990 mean temperature (Marcott et al., 2013), with 1σ uncertainty (gray). c) Timing of deposition of Stalagmite ANJB-2 and MAJ-5. d) Curves representing the sum of glaciers advances from a set of global Holocene time series compiled from natural paleoclimate archives (Wanner et al., 2011). e) curves representing the sum of cold periods from a set of global Holocene time series

compiled from natural paleoclimate archives (Wanner et al., 2011). f) Winter insolation curves
(Berger and Loutre, 1991). G) Summer insolation curves (Berger and Loutre, 1991)

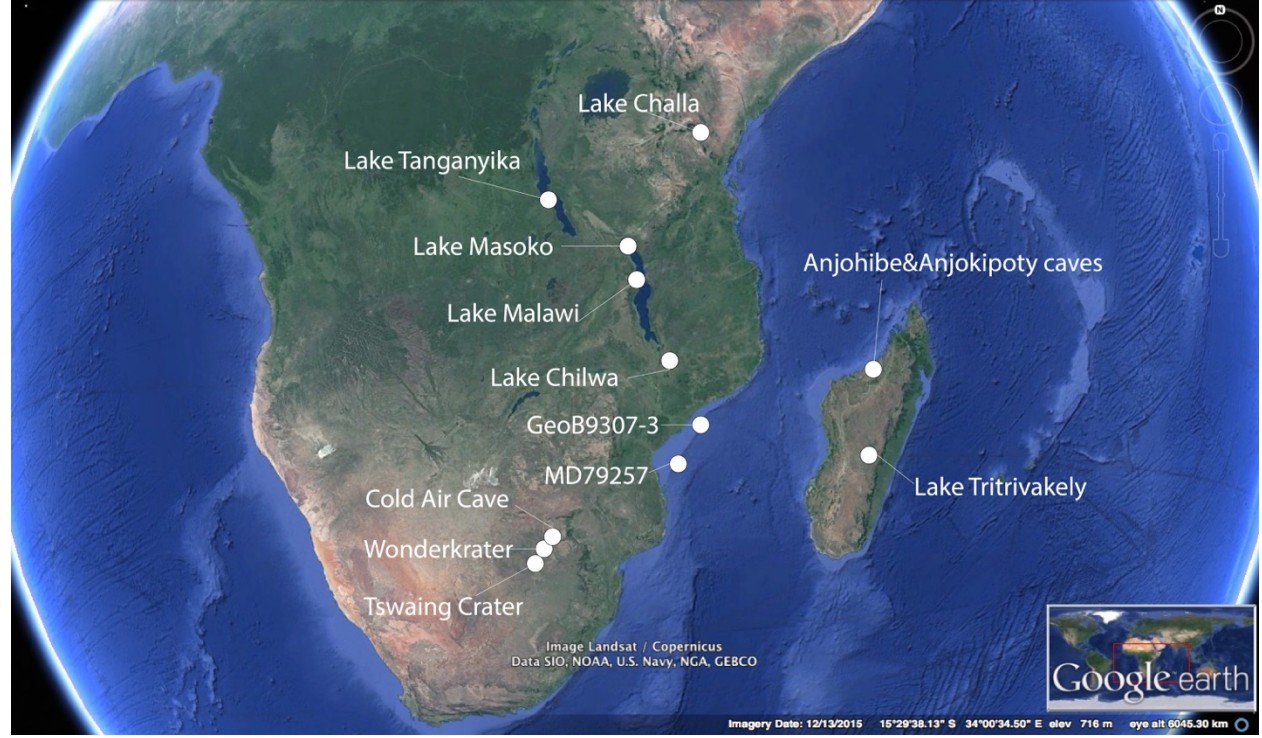



Figure 10: Regional comparison. Google Earth image showing the location of the sites reported
in Table S3 and in Figure 11. Most records reported from these sites are lake sediments, except
for GeoB9307-3 (onshore off delta sediments), MD79257 (alkenone from marine sediment core),
and Cold Air, Anjohibe, and Anjokipoty caves (stalagmites $\delta^{18}$O).

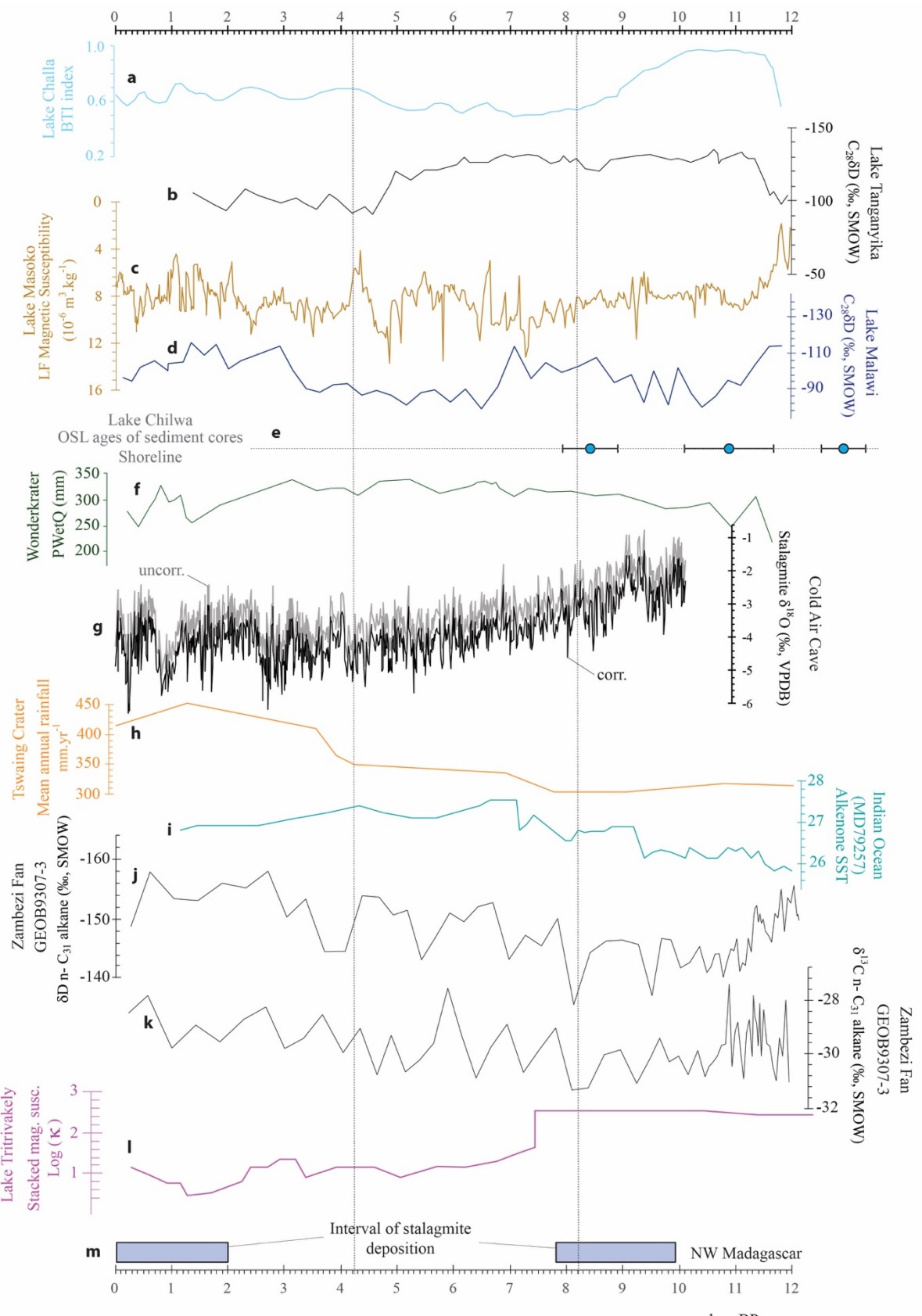


Figure 11: Regional comparison. a) Lake Challa BTI index (Verschuren et al., 2009). b) Lake
Tanganyika $C_{28}$ $\delta$D (Tierney et al., 2008, 2010). c) Lake Masoko low field magnetic susceptibility
($10^{-6}.m^3kg^{-1}$) (Garcin et al., 2006). d) Lake Malawi $C_{28}$ $\delta$D (Konecky et al., 2011). e) Lake Chilwa
OSL dates of shoreline (Thomas et al., 2009). f) Wonderkrater reconstructed paleoprecipitation,
PWetQ (Precipitation of the Wettest Quarter; Truc et al., 2013).  g) Cold Air Cave corrected
(corr.) and uncorrected (uncorr.) $\delta^{18}$O profiles from Stalagmite T8 (Holmgren et al., 2003). h)
Tswaing Crater paleo-rainfall derived from sediment composition (Partridge et al., 1997). i)
Indian Ocean SST records from alkenone (Bard et al., 1997; Sonzogni et al., 1998). j-k) Zambezi
$\delta$D n-$C_{31}$ alkane $\delta^{13}$C n-$C_{31}$ alkane (Schefuß et al., 2011). l) Lake Tritrivakely stacked magnetic
susceptibility (Williamson et al., 1998). m) NW Madagascar (Anjohibe and Anjokipoty) interval of
deposition of Stalagmite ANJB-2 and Stalagmite MAJ-5 (this study). The two vertical dashed lines
indicate the boundary of the Early, Middle, and Late Holocene by Walker et al. (2012) and Head
and Gibbard (2015).

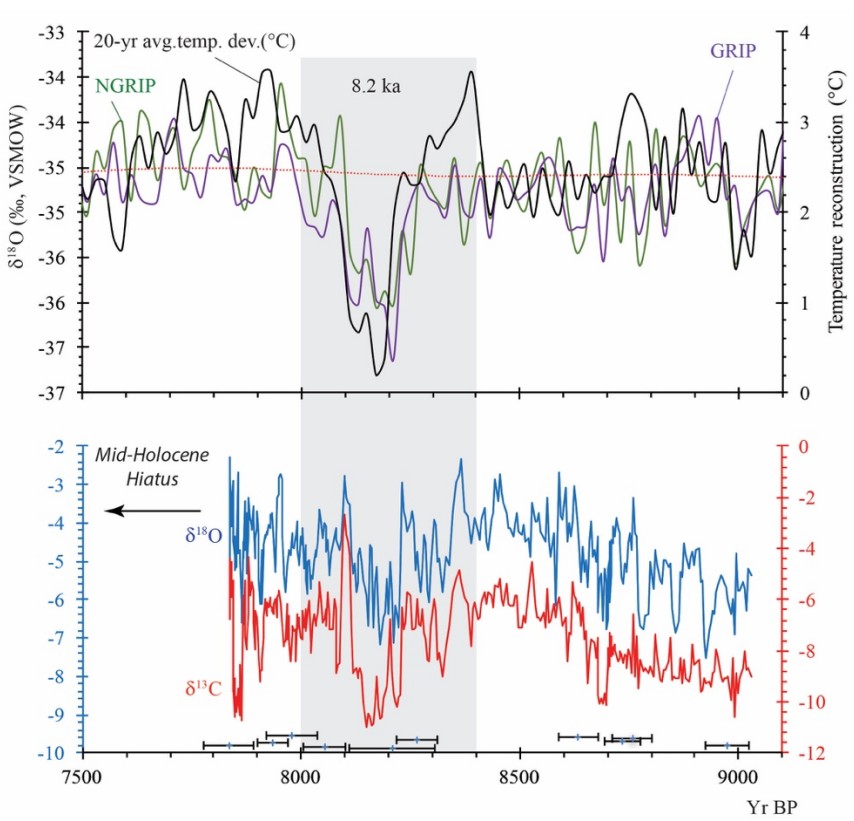


Figure 12: **The 8.2 ka event in Madagascar.** Oxygen isotope record from Greenland (GRIP and
NGRIP) ice cores (Vinther et al., 2009) compared with Stalagmite ANJB-2 $\delta^{18}$O and $\delta^{13}$C.