# Peer review of "Three distinct Holocene intervals of stalagmite deposition and non-deposition revealed in NW Madagascar, and their paleoclimate inferences"

_Climate of the Past, 2016_

## Referee Comment (RC1) · Anonymous Referee #1 · 18 Jan 2017

This study focuses on speleothems from two caves in Madagascar. Several types of analysis are performed including stable isotopes, laminae, and mineralogy, each of which is anchored using U-Th dates. The age models appear robust (although an adequate discussion of age determinations and age model calculations is lacking) but there are several problems. First, the time slices spanned by these stalagmites are quite short, being punctuated by long hiatuses. As a result, the larger context of this record is difficult to identify. Second, I am not convinced of the corrections for differential fractionation between calcite and aragonite d13C values. And associated with this is my concern that there may be microscopically intermingled aragonite and

calcite that can only be corrected for isotopically using quantitative XRD, something that was not done here. Third, replication among samples of the same age is not particularly convincing, raising questions about the controls on isotopic values. Fourth, several claims are poorly substantiated, incompletely referenced, or (to some degree or another) unsupported by the data. Fifth, the writing is at times hard to follow.

1. Does the paper address relevant scientific questions within the scope of CP? Yes 2. Does the paper present novel concepts, ideas, tools, or data? No 3. Are substantial conclusions reached? No 4. Are the scientific methods and assumptions valid and clearly outlined? No 5. Are the results sufficient to support the interpretations and conclusions? No 6. Is the description of experiments and calculations sufficiently complete and precise to allow their reproduction by fellow scientists (traceability of results)? 7. Do the authors give proper credit to related work and clearly indicate their own new/original contribution? Not always 8. Does the title clearly reflect the contents of the paper? No 9. Does the abstract provide a concise and complete summary? 10. Is the overall presentation well structured and clear? No 11. Is the language fluent and precise? No 12. Are mathematical formulae, symbols, abbreviations, and units correctly defined and used? 13. Should any parts of the paper (text, formulae, figures, tables) be clarified, reduced, combined, or eliminated? 14. Are the number and quality of references appropriate? 15. Is the amount and quality of supplementary material appropriate?

Specific comments follow: 18 – is this one cave or two? 23 – why no dates associated with the middle Holocene? 27 – when? 27 - "globally colder" is a little confusing; the interhemispheric temperature gradient is responsible for determining mean global ITCZ position. 30 – when? 33 – is "exemplified" the correct word here? 37 – here is the missing mention of hemispheric temp gradient. I suggest making this explicit earlier in the abstract. 39-40 – delete this sentence 43 – delete "the" 49 – delete "the" 51 – reword as "a particularly" 52 – ITCZ was previously defined 61 – reword "variability of growth-specific width" as "growth laminae" 61 - do not capitalize "cave" 90 – wasn't

replication already discussed on line 42 100 – "long-term" is vague; records of what? 101 – "longer" vague (see previous comment) 112 – "chronologies were" 133 – I am not sure that the correction for carbon isotopic fractionation between calcite and aragonite in speleothems has been adequately explored. As a result, I am uncertain if this part of the results will hold up. 142 – looking at the data table in Supp Materials, it appears that ANJB-2 (sometimes labeled as ANJ-B-2) has a wide range in U abundance. So why the s.d. of 0? 144 – Providing this level of U and Th abundance data is not particularly useful. I would simply refer the reader to the relevant data table. What is missing that should be included here is a discussion of 238/232 ratios in each sample, what 232/232 value was used to correct for inherited 230 (and how this value was derived), and how well the ages fall in correct stratigraphic order. Most ages look quite good but some late Holocene dates have larger errors. These deserve some discussion. 148 – The wording here is confusing. Why argue for some continuous growth intervals but define others as separated by hiatuses? 154 – these are enormous ranges in d18O and d13C. 161 – drop the hundreths place in the stable isotope values (where they are included). It complicates the paper but doesn't have any relevance for interpretation. 205 – this basic introduction should be presented much earlier in the paper if readers who require it are going to glean any meaningful information from the stable isotope results. 272 – relative to what time interval? 276 – I guess, but the record spans so little time that it's hard to get a clear sense of how anomalous this 8.2 isotopic excursion actually is. 278 – "suggest"? The mineralogical composition should be defined precisely (even down to percent calcite or aragonite). Or do you mean to suggest that it may have originally been aragonite but was altered to calcite? 291 – missing a chance to fit this finding into a large context. What other regional records (African, south Asian) record the 8.2 event and what is the nature of these records? 295-297 – I don't understand this sentence. Is this saying what you mean it to say? 373 – there are a lot studies to cite here. I am not sure self-citing is most appropriate in this context. 377 – my reading of much of the SH paleoclimate literature suggests a dominance of NH insolation. 408 – is "he" appropriate useage for Climates of the Past? 416 – similar findings were made

based on lakes and speleothems in South America, and thus it may be worth citing some of this work here. 475 – does the Gulf Stream actually shut down when AMOC slows? Need to cite a modeling stud to support this claim. 729 – is the name for this reference correct? It is a hyphenated name in the text. Fig 5 and Fig 6 – It would be helpful to have the isotopes presented on the same scales oriented along the same horizontal lines so that the reader can assess how each stalagmite's isotopic trends and values compare with the other. Fig 6 – I don't see the connection between solar and stalagmite isotopes here.

---

## Short Comment (SC1) · 12 Feb 2017

This is an important new contribution on the palaeoclimate of Madagascar and the greater southeast African region. The link to the migrating / oscillating ITCZ and the influence of solar activity changes is very important and helps to better understand natural climate variability in the region.

The isotope curves contain additional information which is not fully covered in the discussion section of the paper. For example, I took a closer look at the time of the Medieval Climate Anomaly (1000-1200 AD) and noticed that the Anjohibe Cave records a

general wet phase 850-1100 AD based on d18O. Notably, the d18O development in Anjokipoty Cave differs. Why? A wetter MCA fits well with the bulk of other regional studies from the region (green dots in this regional MCA mapping project: http://t1p.de/mwp ).

It is unfortunate that the two d18O curves in Fig. 5b are plotted on top of each other, making it very hard to see the individual curves. I suggest you separate them for better readability. In the data supplement figure S7 you show datasets AB2 and AB3 without properly introducing them. Please add information on these datasets.

---

## Referee Comment (RC2) · Anonymous Referee #2 · 22 Mar 2017

This papers presents climate reconstruction obtained from two speleothems located in northwestern Madagascar. Three climatic episodes are identified based on change in $\partial^{18}O$ and $\partial^{13}C$. The mid Holocene interval is represented by a hiatus that lasted from 7.8 to 1.6 ka. Petrology, mineralogy and stable isotopes are inferred to discuss changes in stalagmite physiognomy and geochemical composition and relate to climatic changes. The discussion on how to detect hiatuses in speleothems is very interesting with issues of broad interest. However several concerns that are listed below are preventing from allowing a publication of these results in their actual presentation. 1) a discussion on age results and age model is lacking. 2) results show several discrepancies between the two speleothems at a same age which are not commented. A presentation of the curves separately is needed with a discussion on the results. 3) the results are never discussed at a regional scale and some important references are lacking from paleoclimate reconstructions in eastern Africa and Indian Ocean. The Holocene wet-dry-wet succession was already identified in several studies never cited here. The climate boundaries of the Holocene might be spatially limited but not nonexistent. 4) the ITCZ is presented as the main and only driving force to explain the regional hydrological changes ignoring the Indian Ocean Dipole.

Setting Describe the climatic anomalies that are observed today.

Discussion 8.2 ka : all right I can see a decrease in $\partial 18O$ and $\partial 13C$ . However before and after 8.2 k we observe that Âń the similar patterns as the $\partial 18O$ of Greenland Âż is absent. Is similarity only detected at 8.2k ?

The discussion on ITCZ is too long and includes too many generalities nd no novelties. New scientific questions should arise at the end of the paper.

Figures We need a map with the location of the other paleoclimate reconstructions around the Indian Ocean and Eastern Africa

Figure 5 I can see many differences between the two sites at a same age.

Figure 6 I do not understand this figure.

---

## Author Comment (AC1) · 13 Apr 2017

**Revision notes for cp-2016-137**

"Three distinct Holocene intervals revealed in NW Madagascar: evidence from two stalagmites from two caves, and implications for ITCZ dynamics"

by Voarintsoa et al.

**Acknowledgment:**

We thank the CP editor(s) for handling this manuscript.

We thank the reviewers (RC1, SL, and RC2) for their time and efforts at providing comments and suggestions to improve this manuscript.

**Summary:**

- All suggestions by the two anonymous reviewers (RC1 and RC2) and by Sebastian Luening (SL) have been considered carefully. With these regards, the manuscript has been fully revised (major revisions from our side), but since we were only requested to post our responses at this stage, we made sure to list our responses point by point (as detailed as we can) below.
- Comments from all three reviewers suggest revision of many of our figures, and thus Figures 5, 6, and 7 were revised fully (new figures were also added). Below is the list of changes we made about figures:
  - Previous figure 1 has not changed
  - Previous figure 2: we separated the figures so that a new Figure 2 will be showing the age model and a new Figure 3 will be showing the scanned images with the LSW measurements. We did this to make figure presentation better.
  - Previous figure 3: we separated these figures, so that a new Figure 4 shows the scatterplot and two new Figures 5 and 6 show the stable isotope profiles for the Malagasy Early Holocene Interval (MEHI) and the Malagasy Late Holocene

Interval (MLHI), respectively. Presentation of the time series separately will satisfy all reviewers' concern about Figs. 5 and 6.

- Previous figure 4: this figure has not changed except the wet/dry/wet to indicate MEHI-MMHI-MLHI. It's number in the revised manuscript is Figure 7
- Previous figures 5 and 6: These figures have been revised completely, not only to consider the reviewer's suggestions but also to match with the revision in the discussion. We grouped all the possible Holocene forcing (insolation, Temperature, glacier advances and cold periods), pertaining to the ITCZ, into one figure (now Figure 8). The stable isotopes time series were updated earlier.
- A new Figure 9 has been added to consider the regional comparison, as suggested by RC1, but mainly RC2.
- Previous figure 7: This figure has also been revised to zoom the 8.2 ka (i.e. an interval between 7800 and 9100 BP). Its number in the revised manuscript is 10.
- We shortened the discussion on ITCZ as suggested by RC2
- Several additions have been made to the revised manuscript:
  - Table 1 is new and it summarizes regional Holocene climate in eastern Africa and surrounding Indian Ocean (as suggested by RC1, but mainly RC2)
  - Figure 9 (as stated above) is new and it is a map showing the position of all locations described in the regional comparison as discussed in the revised manuscript
  - We added a new section revising the climatic setting and the current anomalies in NW Madagascar and surrounding as requested by RC2
  - a very new section discussing ocean oscillation (mainly IOD, but we touched base on ENSO too) was added in response to RC2's request
- We added several figures in the supplementary documents (these figures are not numbered in this revision note, but they will receive actual numbers when the final version of this manuscript is published):
  - Sketches showing how the LSW (Layer-specific width) was measured
  - o Three figures showing all the X-Ray Diffraction profiles
  - Figures showing selected (micro)photographs to illustrate some of the interpretation (will be added in the revision)
  - Figure showing SEM image of selected aragonite and calcite layer

- Figure showing the stable isotope profile (raw and considering correction for isotopic fractionation)
- The list of references has been fully revised in the revised manuscript
- Detailed responses to RC1 are in pages 4-35
- Detailed responses to Sebastian Luening are in pages 35-37
- Detailed responses to RC2 are in pages 37-49

Reviewers' comments are in black.

Authors' responses are in blue.

Please note that all figures in this revision note (Authors' comments, AC) is not numbered (but they are in the revised manuscript). These figures are incorporated next to the describing/corresponding text.

**Anonymous Referee #1 (RC1):**

**Received and published: 18 January 2017**

Here we provided responses to both the general comments and the specific comments.

This study focuses on speleothems from two caves in Madagascar. Several types of analysis are performed including stable isotopes, laminae, and mineralogy, each of which is anchored using U-Th dates.

**Authors' note:** The reviewer mentioned "laminae" but we'd like to rectify that we used "layer-specific width" (LSW) and not laminae. This LSW method has been implemented by Sletten et al., 2013 (see their Fig. 2) and Railsback et al., 2014 (see their Fig. 3). LSW is the horizontal distance between two points on the flanks, i.e. points at maximum convexity, of a stalagmite. It is the width that exists near the top of the stalagmite at the time when it was deposited. For a given point, LSW is measured on the growth axis of the stalagmite by determining the horizontal distance across the stalagmite between the points at which the corresponding growth surface becomes tangent to a line inclined 35° to the growth axis. Measurements were done at macroscopically traceable layers and plotted as a function of depth. LSW may vary along the stalagmite's growth axis. Narrow LSW suggests drier conditions and wider LSW suggests wetter conditions.

Figure\_: Scheme used to determine layer-specific width (this figure will be added in the supplementary section)

The age models appear robust (although an adequate discussion of age determinations and age model calculations is lacking) but there are several problems.

**Response:** We added a discussion of age determination and age model calculation (please see our response to specific comments below)

First, the time slices spanned by these stalagmites are quite short, being punctuated by long hiatuses. As a result, the larger context of this record is difficult to identify.

**Response:** It is common that stable isotopes are the fundamental and are the most used proxies in paleoclimate reconstruction. In such circumstances, samples with hiatuses are often disregarded.

We however do not share the same point of view with RC1, who states that "the larger context of this record is difficult to identify" because we believe that the replication of a long-term hiatus (~6.5ka) within the same time interval of the MMHI from two separate caves, 16 km apart, is not an artifact. This long-term hiatus is confirmed by radiometric dating, petrography, and mineralogy. The two stalagmites collected from Anjohibe and Anjokipoty caves, indeed, show roughly three distinct intervals of deposition and non-deposition: (1) between 9.8 and 7.8 ka BP, stalagmites grew (2) between 7.8 and 1.6 ka BP, stalagmites stopped growing, and (3) after 1.6 ka BP, stalagmites resumed to grow. We see this as the big picture, and we based the subdivision of the Malagasy Holocene in accord to these three intervals. With the basic knowledge on how stalagmites form, the timing of stalagmite deposition and the non-deposition from these two distant caves could largely reflect changes in paleohydrology in NW Madagascar, and this could be linked to changes in climate. The larger context of the records could be simplified as follow: the timing of deposition could generally suggest wet conditions that sufficiently recharge the cave's hydrology to feed the stalagmites, whereas the timing of non-deposition suggest dry conditions that inhibited stalagmite deposition.

We addressed this concern by revising several sections in the discussion, we have specifically given more attention to the interpretation of the Malagasy mid-Holocene interval.

Second, I am not convinced of the corrections for differential fractionation between calcite and aragonite d13C values. And associated with this is my concern that there may be microscopically intermingled aragonite and calcite that can only be corrected for isotopically using quantitative XRD, something that was not done here.

**Authors' note:** we provided responses to general comment here, but also please see specific response to specific comments (page 22-24).

**Summary response:** Yes, the stalagmites reported in this paper contain both layers of aragonite and layers of calcite as suggested by X-ray diffraction (now we have 15 total XRD profiles from 15 subsamples), microscopic observation of 12 thin sections, and macroscopic observation of the polished surface of the stalagmites. Intermingling between aragonite and calcite at microscopic level exists, but they are not abundant. Because a mixture of aragonite and calcite during sampling could change the stable isotope values in speleothems, I was very careful at extracting sample powders and kept record of the mineralogy each time I drilled a sample. Since the sample size for stable isotopes is very small (1.5 x0.5x0.5 mm), avoiding such mineralogical contamination was possible. To give readers the freedom of interpreting and evaluating the stable isotope data with minimal influence from such correction, we decided to update the stable isotope time series in all the figures of the main manuscript to only show the raw data, and added a separate time series plot in the supplementary document showing both the untransformed and transformed values. We revised the texts in the manuscript accordingly.

Figure\_: Example of the stable isotope of oxygen and carbon profiles showing the raw and corrected values for ANJB-2 during the MEHI.

**Detailed response**:**

*Literature review of C&O in calcite and aragonite that pertains to reviewer's comments:*

The reviewer made a very good point at addressing this aragonite-calcite mixture in stalagmites. Yes, intermingled aragonite-calcite in stalagmite layer could be expected (e.g. Frisia et al., 2002; Gonzalez and Lohmann, 1988; Ortega et al., 2005; Railsback et al., 1994; Woo and Choi, 2006), either as a result of diagenetic processes (e.g. Zhang et al., 2014) or purely primary crystallization of calcite and aragonite when the solution is saturated with these minerals (e.g. Sletten et al., 2013; Railsback et al., 1994). In several aragonite-bearing stalagmite, it is very likely that the aragonite-calcite mixture at microscopic level is expected (e.g. Frisia et al., 2002; Railsback et al., 1994; Lachniet et al., 2012), and this mixture, often variable in proportions (e.g. see Fig. 3 of Sletten et al., 2013; Zhang et al., 2014, Scroxton et al., 2017), could complicate

interpretation of stable isotope variations (Frisia et al., 2002; Zhang et al., 2014; McMillan et al., 2005) in a strict paleoclimate context (Fairchild et al., 2006; Lachniet et al., 2012). The complication specifically arises from the different H2O-CaCO3 equilibrium fractionation factors for aragonite and calcite (e.g. Lachniet, 2009; Rubinson and Clayton, 1969; Romanek et al., 1992; Kim et al., 2007). Investigation of the polished surface of the two stalagmites suggests that the samples did not experience extensive diagenetic alteration, such as the case identified in Zhang et al. (2014) and Lachniet et al. (2012).

We agree that X-ray diffraction is the excellent method to quantify the calcite-aragonite proportion in stalagmite, such as performed by Frisia et al., 2002; Sletten et al., 2013; Zhang et al., 2013; Zhang et al., 2014; Lachniet et al., 2012; Scroxton et al., 2017). We indeed ran ten additional X-ray diffraction analyzes (see figures below, these figures will be added in the supplementary section) on our stalagmites, thus a total of 15 X-ray diffraction analyzes to better resolve the pattern of mineral distribution in our stalagmites. Those X-ray diffraction results agree with our original observation, which was combined with careful microscopic observation of the samples. We specifically looked at twelve oversized thin sections under Leitz Laborlux 12 and under Leica DMLP, equipped with QCapture, to have a thorough understanding of the sample's internal structure, texture, and mineralogy. We have found that the boundary between calcite and aragonite throughout the sample is overall sharp (see figure below). Above all these, we did not find strong evidence of diagenetic alteration.

---

## Author Response (AR1)

**Revision notes no. 2 for cp-2016-137**

**2017.07.10**
* * *
Editors and Reviewers notes are indicated in black in this note.

Authors comments are in blue. All changes made in the manuscript are also in blue (we added in the comment sections some specific response to RC1, RC2, SL, and EC)

RC1 and RC2: Reviewer #1 and #2, respectively

SL: Sebastian Luening

EC: Editor's comments/Editor
* * *
We would like to note that the manuscript has been thoroughly revised (major revision), and thus some comments/suggestions of changes by RC are no longer applicable (in fact, several sentences were removed). However, we tried our best to refer Editors and Reviewers to sections that correspond to the requested changes.

*Although we indicated specific lines to address the comments, we kindly invite editor and reviewers to read the manuscript and its accompanying supplementary documents anew. Thank you for considering this manuscript.*
* * *
**Major changes:**

1) After fully revising the submitted manuscript, its title has been updated to: "*Three distinct Holocene intervals of stalagmite deposition and non-deposition revealed in NW Madagascar, and their paleoclimate inferences*"

2) The abstract has been fully revised to reflect the changes made to the manuscript

3) Many figures were removed and updated to address major comments from RC1, RC2, and EC.

4) Here is a list of major changes in response to RC1, RC2, EC (detailed responses are given further below):

   a. Radiometric dating (please see lines 180–198 and lines 256–284)

   b. Stable isotopes (please see lines 221–253 and 287–309), also please see Figs 5–6 and their corresponding supplementary Figures S6–S8.

c.  Most of the figures have been fully revised as indicated in the Author's Responses to Reviewers and Editor d.  Some subsections have been added to clarify ideas (e.g., Sect. 3. Methods, now with three distinct subsections)

e.  Interpretation of the three intervals of the Holocene in Madagascar has been revised (Sect. 5.2. Lines 398–524)

f.  The section 5.3. on the ITCZ implications has been shortened (Lines 505-524) in response to RC2

g.  New sections have been added, and they are:

    i.  Sect. 2.1. Stalagmites and their setting (lines 76–98): this section was moved as suggested by RC1

    ii.  Sect. 2.3. Climate of Madagascar: new and inserted in response to RC2

    iii.  Sect. 5.4. Regional comparison, added in response to RC2

    iv.  Sect. 5.6. Beyond the ITCZ: IOD and ENSO influence on Madagascar's climate (Lines 576–605), added in response to RC2

h.  Several changes have been made to the Supplementary materials (in fact we added several figures and texts)
* * *
**Specific responses to Reviewers and Editors**

**Editor:**

In addition to responding to the reviewer comments I would also ask that you pay particular attention to:

*detailing the mineralogical assignment of the speleothems and how this leads to the isotopic correction along the length of the samples. In particular it is unclear how each isotopic sample is corrected for it's aragonite:calcite composition when only a small number of discrete XRD analyses have been completed.

•  Please see specifically Lines 228–229 and 239–240

•  Also, please see Lines 200–241

- (The excel file with the data were color coded to account for the difference in mineralogy, and thus to ease mathematical correction for stable isotopes). Those data will be made available publicly and submitted to NOAA upon acceptance of this manuscript.
- Additional explanation: we run a total of 15 XRD samples to identify the nature of mineralogy with similar fabrics. For the isotopic correction, the mineralogy at the crest was mostly monomineralic, but we specified the correction at lines 239–240. (If this is very confusing, I am happy to discuss this via skype if necessary).

*detailing the U-series and age model details. In particular more detail needs to be given on initial 230/232 values and the effect that the chosen values have on bringing U-series ages into stratigraphic alignment. Age information also needs to be included on plots showing the isotopic time-series so that readers can evaluate how much "movement" of the records is possible based on age uncertainty.

- Please see Lines 180–198 and Lines 256–277
- Age information: please see Figs. 5–6

*be careful with "wiggle matching". The examples in the proposed revised figures are not convincing to me. Please be careful in how you approach this and in how robust your climatic interpretations are.

- We removed all wiggle matching to avoid biases in interpretation.

*more thoroughly compare your data with other records from the region (not just a map of the locations of other sites). The Scroxton et al 2017 QSR paper that also presents speleothem data from Madagascar should also be included in the revised interpretation of your records.

- Done, please see Figure 11 (also see Sect. 5.4 at Lines 527–574)
- For Scroxton et al. (2017), please see Lines 127 and 579–580

**Anonymous Referee #1 (RC1):**

This study focuses on speleothems from two caves in Madagascar. Several types of analysis are performed including stable isotopes, laminae, and mineralogy, each of which is anchored using U-Th dates.

- We added a paragraph to indicate that we used Layer-Specific Width (LSW) and not laminae. Please see lines 211–218.

The age models appear robust (although an adequate discussion of age determinations and age model calculations is lacking) but there are several problems.

- please see lines 180–198 and lines 256–284

First, the time slices spanned by these stalagmites are quite short, being punctuated by long hiatuses. As a result, the larger context of this record is difficult to identify.

- To specifically address this, please see Lines 278–284
- We also revised the interpretation of the three intervals of the Holocene (Lines 398–524)

Second, I am not convinced of the corrections for differential fractionation between calcite and aragonite d13C values. And associated with this is my concern that there may be microscopically intermingled aragonite and calcite that can only be corrected for isotopically using quantitative XRD, something that was not done here.

- Please see specifically Lines 228–229 and 239–240
- Also, please see Lines 200–241
- Figures S9–S11
- (The excel file with the data were color coded to account for the difference in mineralogy, and thus to ease mathematical correction for stable isotopes). Those data will be made available publicly and will submitted to NOAA upon acceptance of this manuscript.

Third, replication among samples of the same age is not particularly convincing, raising questions about the controls on isotopic values.

- Please see Lines 301–309

Fourth, several claims are poorly substantiated, incompletely referenced, or (to some degree or another) unsupported by the data.

- The manuscript has been fully revised

Fifth, the writing is at times hard to follow.

- We did our best to carefully revise the manuscript. We incorporated requested changes in the specific comments by RC1 below. We also incorporated changes according to RC2's comment (e.g. narrowing discussion of the ITCZ and inserting discussion on other climate forcings, like IOD). Please see our responses to specific comments and please see our response to RC2.

1. Does the paper address relevant scientific questions within the scope of CP? Yes
2. Does the paper present novel concepts, ideas, tools, or data? No
   - Please see Authors' responses
3. Are substantial conclusions reached? No
   - Please see Authors' responses
4. Are the scientific methods and assumptions valid and clearly outlined? No
   - Please see Sect. 3 Methods (with three new subsections to make this clearer) at lines 179–253
5. Are the results sufficient to support the interpretations and conclusions? No
   - Please see Authors' responses
6. Is the description of experiments and calculations sufficiently complete and precise to allow their reproduction by fellow scientists (traceability of results)?
7. Do the authors give proper credit to related work and clearly indicate their own new/original contribution? Not always 8. Does the title clearly reflect the contents of the paper? No
   - Please see Authors' responses
9. Does the abstract provide a concise and complete summary?

10. Is the overall presentation well structured and clear? No

- Please see Authors' responses

11. Is the language fluent and precise? No

- Please see Authors' responses

12. Are mathematical formulae, symbols, abbreviations, and units correctly defined and used?

13. Should any parts of the paper (text, formulae, figures, tables) be clarified, reduced, combined, or eliminated?

14. Are the number and quality of references appropriate?

15. Is the amount and quality of supplementary material appropriate?

Specific comments follow:
* * *
The comments below no. 18–40 belong to the Abstract. We fully revised the Abstract to account for the changes made in the manuscript, thus responses to specific comments no. 18–40 become N/A.

– is this one cave or two?

– why no dates associated with the middle Holocene?

– when?

- "globally colder" is a little confusing; the interhemispheric temperature gradient is responsible for determining mean global ITCZ position.

– when?

– is "exemplified" the correct word here?

– here is the missing mention of hemispheric temp gradient. I suggest making this explicit earlier in the abstract.

39-40 – delete this sentence

– delete "the"

- deleted

– delete "the"

- deleted

– reword as "a particularly"

- sentence revised (Lines 58–59)

– ITCZ was previously defined

- In the previous version of the manuscript, the ITCZ was defined in the abstract. In the revised manuscript, we defined the first acronym at the beginning in the introduction, and used the ITCZ acronym for the remainder of the text. (Lines 56–57)

– reword "variability of growth-specific width" as "growth laminae"

- We already explained this above (LSW)

- do not capitalize "cave"

- corrected

– wasn't replication already discussed on line 42

- Section revised (Lines 101–117), repeated ideas/texts were removed.

– "long-term" is vague; records of what?

- Please see lines 173–176

– "longer" vague (see previous comment)

- see our response to line 100 (above)

– "chronologies were"

- Corrected (Lines 193)

– I am not sure that the correction for carbon isotopic fractionation between calcite and aragonite in speleothems has been adequately explored. As a result, I am uncertain if this part of the results will hold up.

- Please see Sect. 3.3. (Lines 220–241)

– looking at the data table in Supp Materials, it appears that ANJB-2 (sometimes labeled as ANJ-B-2) has a wide range in U abundance. So why the s.d. of 0? 144 – Providing this level of U and Th abundance data is not particularly useful. I would simply refer the reader to the relevant data table. What is missing that should be included here is a discussion of 238/232 ratios in each sample, what 232/232 value was used to correct for inherited 230 (and how this value was derived), and how well the ages fall in correct stratigraphic order. Most ages look quite good but some late Holocene dates have larger errors. These deserve some discussion.

- Considered, please see lines 257–277
- The labels were also corrected (thank you!)

– The wording here is confusing. Why argue for some continuous growth intervals but define others as separated by hiatuses?

- This section was revised, please see lines 278–284

– these are enormous ranges in d18O and d13C. 161 – drop the hundreths place in the stable isotope values (where they are included). It complicates the paper but doesn't have any relevance for interpretation.

- Considered, please see lines 287–293 and 294–300 (we also revised these paragraphs)

– this basic introduction should be presented much earlier in the paper if readers who require it are going to glean any meaningful information from the stable isotope results.

- Done, and moved to Sect. 2.1 (Lines 76–98)

– relative to what time interval?

- Relative to modern climate (Lines 402)

– I guess, but the record spans so little time that it's hard to get a clear sense of how anomalous this 8.2 isotopic excursion actually is.

- Instead of arguing that this is anomalous, and to avoid over-interpretation of the data, we revised the sentence (please see Lines 406–407) and its corresponding figure 12.

– "suggest"? The mineralogical composition should be defined precisely (even down to percent calcite or aragonite). Or do you mean to suggest that it may have originally been aragonite but was altered to calcite?

- Please see lines 406–416

– missing a chance to fit this finding into a large context. What other regional records (African, south Asian) record the 8.2 event and what is the nature of these records?

- We dedicated an updated section on the 8.2 ka (Lines 549–574)

295-297 – I don't understand this sentence. Is this saying what you mean it to say?

- We deleted as we revised the manuscript

– there are a lot studies to cite here. I am not sure self-citing is most appropriate in this context.

- Since the sections on ITCZ have been shortened (in response to RC2), that sentence was deleted. However, please see Sect. 5.3 (Lines 505–524)

– my reading of much of the SH paleoclimate literature suggests a dominance of NH insolation.

- Since the sections on ITCZ have been shortened (in response to RC2), that sentence was deleted. However, please see Sect. 5.3 (Lines 505–524)

– is "he" appropriate useage for Climates of the Past?

- Sentence deleted after major revision

– similar findings were made based on lakes and speleothems in South America, and thus it may be worth citing some of this work here.

- See lines 517–524

– does the Gulf Stream actually shut down when AMOC slows? Need to cite a modeling stud to support this claim.

- The section on the 8.2 ka was fully revised (Lines 549–574)

– is the name for this reference correct? It is a hyphenated name in the text.

- Good eyes! The reference is now corrected. (Line 1106)

Fig 5 and Fig 6 – It would be helpful to have the isotopes presented on the same scales oriented along the same horizontal lines so that the reader can assess how each stalagmite's isotopic trends and values compare with the other.

Fig 6 – I don't see the connection between solar and stalagmite isotopes here.

For any comments pertaining to figures, we revised many of them (as listed in the Authors' Responses summary submitted earlier). We kindly invite reviewers to look at the new Figures and the supplementary materials.

**Short Comment** *by* Sebastian Luening (**SL**)

luening@uni-bremen.de

This is an important new contribution on the palaeoclimate of Madagascar and the greater southeast African region. The link to the migrating / oscillating ITCZ and the influence of solar activity changes is very important and helps to better understand natural climate variability in the region.

Thank you!

The isotope curves contain additional information which is not fully covered in the discussion section of the paper. For example, I took a closer look at the time of the Medieval Climate Anomaly (1000-1200 AD) and noticed that the Anjohibe Cave records a general wet phase 850-1100 AD based on d18O.

- Please see lines 484–488

Notably, the d18O development in Anjokipoty Cave differs. Why?

- Please see Lines 301–309

A wetter MCA fits well with the bulk of other regional studies from the region (green dots in this regional MCA mapping project: http://t1p.de/mwp).

It is unfortunate that the two d18O curves in Fig. 5b are plotted on top of each other, making it very hard to see the individual curves. I suggest you separate them for better readability.

- Figures were revised

In the data supplement figure S7 you show datasets AB2 and AB3 without properly introducing them. Please add information on these datasets.

- The manuscript has been fully revised, and this figure is no longer needed, thus it was deleted.

**Anonymous Referee #2 (RC2):**

This papers presents climate reconstruction obtained from two speleothems located in northwestern Madagascar. Three climatic episodes are identified based on change in $\partial$ 18O and $\partial$13C. The mid Holocene interval is represented by a hiatus that lasted from 7.8 to 1.6 ka. Petrology, mineralogy and stable isotopes are inferred to discuss changes in stalagmite physiognomy and geochemical composition and relate to climatic changes. The discussion on how to detect hiatuses in speleothems is very interesting with issues of broad interest. However several concerns that are listed below are preventing from allowing a publication of these results in their actual presentation.

1) a discussion on age results and age model is lacking.

- please see lines 180–198 and lines 256–284

2) results show several discrepancies between the two speleothems at a same age which are not commented. A presentation of the curves separately is needed with a discussion on the results.

- Figures were separated, thank you for this suggestion!
- Discrepancies discussed in text (please see Lines 301–309)

3) the results are never discussed at a regional scale and some important references are lacking from paleoclimate reconstructions in eastern Africa and Indian Ocean. The Holocene wet-dry-wet succession was already identified in several studies never cited here. The climate boundaries of the Holocene might be spatially limited but not nonexistent.

- New section on regional comparison was added (please see Sect. 5.4. at Lines 526–547)

4) the ITCZ is presented as the main and only driving force to explain the regional hydrological changes ignoring the Indian Ocean Dipole.

- New section on other factors than ITCZ was added (please see Sect. 5.6. at Lines 576–605)

Setting Describe the climatic anomalies that are observed today.

- Please see Sect. 2.3. at Lines 119–164. This is a new section added to the manuscript.

Discussion 8.2 ka: all right I can see a decrease in $\partial$ 18O and $\partial$13C . However before and after 8.2 k we observe that Ân´ the similar patterns as the $\partial$18O of Greenland Âz˙ is absent. Is similarity only detected at 8.2k ?

- Since the 8.2 represents an interval of interest of the early Holocene, we focused on its interpretation (section 5.5 was revised)

The discussion on ITCZ is too long and includes too many generalities nd no novelties. New scientific questions should arise at the end of the paper.

- Considered (the section on ITCZ has been shortened, please see Sect. 5.3. at lines 505–524)
- New critical points added at the end of the paper (Lines 621–629). Good suggestion, thank you!

Figures We need a map with the location of the other paleoclimate reconstructions around the Indian Ocean and Eastern Africa

- Done, and we added another figure (Fig. 11) showing comparison of time series (as suggested by EC)

Figure 5 I can see many differences between the two sites at a same age.

Figure 6 I do not understand this figure.

- We revised several figures, including figure 5 and 6. We kindly invite RC2 to have a look at the revised figures and the supplementary figures. Thank you!

**Three distinct Holocene intervals of stalagmite deposition and non-deposition revealed in NW Madagascar, and their paleoclimate inferences**

Voarintsoa, Ny Riavo G.[1*], L. Bruce Railsback[1], George A. Brook[2], Lixin Wang[2], Gayatri Kathayat[3], Hai Cheng[3,4], Xianglei Li[3], R. Lawrence Edwards[4], Rakotondrazafy Amos Fety Michel[5], Madison Razanatseheno Marie Olga[5]

[1] Department of Geology, University of Georgia, Athens, GA 30602-2501 U.S.A.
[2] Department of Geography, University of Georgia, Athens, Georgia, 30602-2502 U.S.A.
[3] Institute of Global Environmental Change, Xi'an Jiaotong University, Xi'an, Shaanxi 710049, P.R. China
[4] Department of Earth Sciences, University of Minnesota, Minneapolis, Minnesota 55455, U.S.A.
[5] Department of Geology, University of Antananarivo, Madagascar

*Correspondence to: Ny Riavo Voarintsoa (nv1@uga.edu or nyriavony@gmail.com)

ABSTRACT

Petrographic features, mineralogy, and stable isotopes from two stalagmites collected from Anjohibe and Anjokipoty caves allow distinction of three intervals of the Holocene in NW Madagascar. The Malagasy early Holocene (between c. 9.8 and 7.8 ka) and late Holocene (after c. 1.6 ka) intervals (MEHI and MLHI, respectively) record evidence of stalagmite deposition. The Malagasy middle Holocene interval (MMHI, between c. 7.8 ka and 1.6 ka), however, is marked by a depositional hiatus lasting for c. 6500 years.

Deposition of Stalagmites ANJB-2 and MAJ-5 from Anjohibe and Anjokipoty caves, respectively, during the MEHI and the MLHI suggests that these caves were sufficiently supplied with water to allow stalagmite formation. These MEHI and MLHI intervals may have been comparatively wet. In contrast, the long-term depositional hiatus likely suggests that the MMHI was relatively drier than the MEHI and the MLHI. This dry condition could have influenced the amount of water supplied to the cave, and thus prevented formation of the stalagmites.

The alternating "wet/dry/wet" during each of these Holocene intervals could be generally linked to the long-term migration of the Inter-Tropical Convergence Zone (ITCZ). When the ITCZ's mean position is farther south, NW Madagascar experiences wetter conditions, such as during the MEHI and MLHI, and when it moves north, NW Madagascar climate becomes drier, such as during

Comment [NRG1]: This abstract has been fully revised to reflect the major changes done to the manuscript 
[revised manuscript text omitted]

**Comment [NRG3]:** Revised in response to RC1. Some idea repetitions were removed

*2.3.* Climate of Madagascar

Comment [NRG4]: New section inserted in response to RC2

[revised manuscript text omitted]

**Comment [NRG7]:** In this method section, we clearly added a subheading indicating the different approaches performed in this study

**Comment [NRG8]:** This section has been revised to address RC1 and RC2's request about details about radiometric dating
We added information about the correction as requested by RC1

of Georgia, 3) by  using scanning electron microscopy (SEM) to better understand the mineralogical fabrics at locations of interest (Fig. S13), and 4) by analyzing about 30–100 mg of powdered spelean layers (n=15) on a Bruker D8 X-ray Diffractometer in the Department of

Geology, University of Georgia. For calcite and aragonite identification, we used CoKα radiation at a 2θ angle between 20° and 60°.

Layer-specific width (LSW) of clearly-defined layers was measured at selected locations on the stalagmite polished surfaces (Fig. S4; Sletten et al., 2013; Railsback et al., 2014; Voarintsoa et al., 2017b). LSW is the horizontal distance between two points on the flanks of the stalagmite where convexity is greatest. It is the width near the top of the stalagmite when the layer being examined was deposited. LSW is measured at right angles to the growth axis of the stalagmite; it is the horizontal distance between points on the layer growth surface becomes tangent to a line inclined at 35° to the growth axis (Fig. S4). LSW may vary along the length of the stalagmite, with smaller values suggesting drier conditions and larger values wetter conditions.

3.3. Stable isotopes

Stable isotope samples of 50–100 µg were manually drilled along the stalagmite's growth layers at the crest. The trench size is very small (1.5 x 0.5 x 0.5 mm). Since a small mixture of calcite and aragonite could potentially change the $\delta^{18}$O and $\delta^{13}$C of the measured spelean layers (see for example Frisia et al., 2002), drilling and sample extraction was carefully done on individually discrete layers using the smallest drill-bit head (SSW-HP-1/4) to avoid potential mixing between calcite and aragonite. The polished surface of the two stalagmites were examined to see if features of diagenetic alteration are present (see for example fig. 2 of Zhang et al., 2014), but none was found. During sampling, the mineralogy at the crest, where stable isotope samples were extracted, was recorded for future mineralogical correction.

Aragonite oxygen and carbon isotopic corrections were performed to compensate for aragonite's inherent fractionation of heavier isotopes (e.g., Romanek et al., 1992; Kim et al., 2007;

McMillan et al., 2005) and to remove the mineralogical bias in isotopic interpretation between calcite and aragonite. The correction consists of subtracting 0.8‰ for $\delta^{18}$O (Kim and O'Neil, 1997;

Tarutani et al., 1969; Kim et al., 2007; Zhang et al., 2014) and 1.7 ‰ for $\delta^{13}$C (Rubinson and Clayton,

**Comment [NRG9]:**
Re: isotopic correction and mineralogy

In response to RC1 and EC, we run more X-Ray diffraction (see supplementary figures). Fifteen samples were run, which approximate XRD analyses done in other studies (e.g., Zhang et al., 2013; Zhang et al., 2014; Scroxton et al., 2017) X-ray diffraction helped at understanding the nature of mineralogy with similar texture and fabric, and thus running XRD for each stable isotope subsample is unnecessary. It is also technically unfeasible given the larger sample size (a minimum of 30 mg of powder for one XRD versus 50-100 micrograms for stable isotopes).

We, however, carefully recorded the mineralogy at each crest while extracting subsamples for stable isotopes. This information is crucial to correct for isotopic values.

The isotopic correction was done based on the mineralogy at the crest where stable isotopes were extracted.

**Comment [NRG10]:** In this section, we addressed the issue raised by RC1 and EC about isotopic correction. We also added detailed text in the supplementary document.

**Comment [NRG11]:** At each sampling, I recorded the mineralogy of the stalagmite. We run X-ray diffraction to understand the nature of mineralogy by similar texture and mineral fabrics.
(to further address RC1 and EC1 concern)

1969; Romanek et al., 1992) for the aragonite as has been done previously (e.g., Holmgren et al.,

2003; Sletten et al., 2013; Liang et al., 2015; Railsback et al., 2016; Voarintsoa et al., 2017a) as shown in equations 2 and 3 below (where $R_{A/C}$ is the aragonite percentage if not 100%).

$\delta^{18}O_{corr.}$ (‰, VPDB) = $\delta^{18}O_{uncorr.}$ (‰, VPDB) − [$R_{A/C}$ x 0.8 (‰, VPDB)] (Eq. 2)

$\delta^{13}C_{corr.}$ (‰, VPDB) = $\delta^{13}C_{uncorr.}$ (‰, VPDB) − [$R_{A/C}$ x 1.7 (‰, VPDB)] (Eq. 3)

Supplementary Figures S6–S8 show both the corrected and uncorrected isotopic records.

For the analytical methods, oxygen and carbon isotope ratios were measured using the

Finnigan MAT-253 mass spectrometer fitted with the Kiel IV Carbonate Device of the Xi'an Stable

Isotope Laboratory in China (ANJB-2; n=654) and using the Delta V Plus at 50°C fitted with the

GasBench-IRMS machine of the Alabama Stable Isotope Laboratory in USA (MAJ-5; n=286).

Analytical procedures using the MAT 253 are identical to those described in Dykoski et al. (2005), with isotopic measurement errors of less than 0.1 ‰ for both $\delta^{13}C$ and $\delta^{18}O$. Analytical methods and procedures using the GasBench-IRMS machine are identical to those described in Skrzypek and Paul (2006), Paul and Skrzypek (2007), and Lambert and Aharon (2011), with ±0.1 ‰ errors for both $\delta^{13}C$ and $\delta^{18}O$. In both techniques, the results are reported relative to Vienna PeeDee

Belemnite (VPDB) and with standardization relative to NBS19. An inter-lab comparison of the isotopic results was conducted, and it involved replicating every tenth sample of Stalagmite MAJ-

5 at both labs. This exercise showed a strong correlation between the lab results (Fig. S5).

**4. Results**

**4.1. Radiometric data**

Results from radiometric analyses of the two stalagmites are presented in Tables S1 and

S2. Corrected $^{230}$Th ages suggest that Stalagmite ANJB-2 was deposited between c. 8977±50 and c. 161±64 yr. BP, and Stalagmite MAJ-5 was deposited between c. 9796±64 and c. 150±24 yr. BP.

These ages collectively indicate stalagmite deposition at the beginning (between 9.8 and 7.8 ka

BP) and at the end of the Holocene (after c. 1.6 ka BP). In both stalagmites, the older ages have small 2σ errors and they generally fall in correct stratigraphic order, except sample ANJB-2-120

and its replicate ANJB-2-120R, which were not used because of the sample's high porosity and high detrials content. In contrast, many of the younger ages have larger uncertainties.  This is

Comment [NRG12]: This is specifically how the corrections were done (in response to RC1, L133, and EC)

Note: To give readers the freedom of interpreting and evaluating the stable isotope data with minimal influence from such correction, we decided to update the stable isotope time series in all the figures of the main manuscript to only show the raw data, and added a separate time series plot in the supplementary document showing both the untransformed and transformed values.

We also added additional text in the supplementary materials

Comment [NRG13]: As suggested by RC1, we simply refer the reader to the relevant data table (L142)
We discussed about the corrections (which are also detailed in the method section) and about the quality of the data.

RC2 states that a discussion on age results and age model is lacking, here we added details, and we also included more details in the method section mainly because many of the younger samples have very low uranium concentration and the detrital thorium concentration is also high, similar to what Dorale et al. (2004) reported. We also understand that the value for initial $^{230}$Th correction, i.e. the initial $^{230}$Th/$^{232}$Th atomic ratio of 4.4

± 2.2 × 10$^{-6}$ for a bulk earth with a $^{232}$Th/$^{238}$U value of 3.8, in these samples could have slightly altered the $^{230}$Th age of these younger samples, leading to larger uncertainties (such as discussed in Lachniet et al., 2012). We encountered similar problems while working on other younger samples from the same cave, but we compared the stable isotope profile with other published records using isochron corrections, and results did not differ significantly (see Fig. 9 of Voarintsoa et al., 2017b). Since this work does not focus on decadal or centennial interpretation of the Late

Holocene stable isotope data, additional chronology adjustment has not been made, and we used the chronology from StalAge to construct the time series. However, in Figures 5 and 6, age uncertainties are given below the stable isotope profiles so that comparisons with other records can accommodate these uncertainties.

The key finding from our age and petrographic data for the two stalagmites is that they suggest that there were three distinct intervals of growth and non-growth during the Holocene (Figs. 2–4, 7). The information suggesting this includes: (1) CaCO$_3$ deposition between c. 9.8 and

7.8 ka B.P., (2) a long depositional hiatus between c. 7.8 and 1.6 ka B.P., and (3) resumption of

CaCO$_3$ deposition after c. 1.6 ka B.P. In the rest of the paper, we will refer to these intervals as the

Malagasy Early Holocene Interval (MEHI), Malagasy Mid-Holocene Interval (MMHI), and Malagasy

Late Holocene Interval (MLHI), respectively.

**4.2. Stable isotopes**

Raw values of $\delta^{18}$O and $\delta^{13}$C for Stalagmite ANJB-2 range from −8.9 to −2.3‰ (mean = −

5.0‰), and from −11.0 to +5.2‰ (mean = −4.2‰), respectively, relative to VPDB. Raw values of

$\delta^{18}$O and $\delta^{13}$C for Stalagmite MAJ-5 range from −8.8 to −0.9‰ (mean = −4.9‰), and from −9.4 to

+2.6‰ (mean = −4.4‰), respectively, relative to VPDB. Mean $\delta^{18}$O and $\delta^{13}$C values are distinguishable between the MEHI and the MLHI. In both stalagmites, the amplitude of $\delta^{18}$O

fluctuations was fairly constant throughout the Holocene; whereas the $\delta^{13}$C profile shows a dramatic shift toward higher values (i.e. from -10.9‰ to +3.8‰, VPDB) at c. 1.5 ka BP.

**Comment [NRG14]:** RC1 stated that the "*larger context of this record is difficult to identify*". This paragraph would summarize and clarify RC1's concerns. Please also see the discussion section on these three intervals (5.2 and 5.3 at lines 386–502 and lines 504–523)
Also, RC1 was confused with the wording (L148) so we rewrote this paragraph to make it clearer.

**Comment [NRG15]:** All the values in here are revised in response to RC1 (L154)

[revised manuscript text omitted]

Growth and non-growth of stalagmites depends on several factors linked to water availability, which is largely determined by climate (more water during warm/rainy seasons and less water during cold/dry seasons). Water is the main dissolution and transporting agent for most chemicals in speleothems. Cave hydrology varies significantly over time in response to climate, and this variability influences the formation or dissolution of $CaCO_3$. In this regard, calcium carbonate does not form if there is little or no water entering the cave, or if there is too much (see

**Comment [NRG17]:** We moved a paragraph introducing stalagmite in the setting section in response to RC1 (L205)

We however kept the information pertaining to the "Paleoclimate significance of stalagmite and non-growth: implications for paleohydrology" here at the beginning of the discussion section to remind the readers that the timing of deposition and non-deposition of these stalagmites could be the primary key to understand the overall paleohydrology during the Holocene in the studied region.

[revised manuscript text omitted]

**Comment [NRG20]:** Inserted to address RC1 about diagenetic alteration. Three additional figures were added in the supplementary materials

*5.2.2.*  Malagasy mid-Holocene interval (c. 7.8–1.6 ka BP)

The only data we have for the MMHI is the long term (~6.5 ka) depositional hiatus in both stalagmites (Figs. 2–3), that potentially indicate dry conditions. The question is why did neither stalagmite grow during the MMHI? Here, we try to explain the factors and the climatic conditions that may have been responsible for it.

[revised manuscript text omitted]

**Comment [NRG24]:** Inserted in response to RC1 (L416)

**Comment [NRG25]:** This is a new section inserted in response to RC2, a supplementary Table has been added

**Comment [NRG26]:** We added more modeling citations here in response to RC1 (L475)

[revised manuscript text omitted]

---

## Author Response (AR2)

**Revision notes no. 3 for cp-2016-137**

**2017.10. 12**

Editors and Reviewers notes are indicated in black in this note.

Authors comments are in red. All changes made in the manuscript are also in red. Specific responses to Editor's comment are indicated with Line numbers to ease tracking of changes.

We thank the Editor for her time and efforts.

Editor Decision: Reconsider after major revisions (30 Aug 2017) by Nerilie Abram

Comments to the Author:

Thank you for submitting your revised manuscript to Climate of the Past. The manuscript required major revisions and has been substantially improved following the advice of the two reviewers and the short comment made during the discussion phase. Rather than resend your manuscript to the original reviewers I have carried out a detailed review of the manuscript myself. There are still further revisions that I feel are needed before the manuscript could be accepted for publication in Climate of the Past. These are:

*Abstract: the abstract is much too long. Please revise so that the abstract provides a concise account of the motivation and findings of your study.
Revised and shortened (now 239 words) to reflect the updates made in the revised manuscript (Lines 21–39)

*Please make sure that there is consistency in your use of age abbreviations throughout the text. Make sure all abbreviations are defined on first use (e.g. ka) Avoid jumping between different age scales (e.g. ka BP, yr BP, cal BP and AD are all used in section 5.2.3).
Considered. Also, Section 5.2.3 was revised and un-necessary sentences were removed. (now at Lines 440–450)
Definition of ka first appears in section 2.4 at Line 141.

*Please try to reduce the length of the text wherever possible, and in particular keep the text more focused on the processes/concepts that are important for this study. E.g. There are long discussions about IOD and ENSO in both sections 2.3 and section 5.6, but no interpretations are made regarding the role that these climate processes may have played in the climate of Madagascar during the Holocene, so these sections should be much shorter than they currently are, or incorporated better into the interpretations made with your records.
Section 2.3. was revised and shortened (now at Lines 108–133), and Section 5.6 was moved to supplementary (now in the Suppl. Text no. 4).

*There are some sections that are still far too speculative and should be removed from the text. Things that in my opinion can't be justified include:

- the discussion in section 5.2.3 linking events in the last millennium with the speleothem isotope records. The lack of replication of the isotopic records between the two samples makes this conclusion very weak. I would suggest sticking just with the broader-scale "growth-non growth" information, rather than trying to interpret features of the poorly reproduced isotope records.
This section has been revised (Lines 440–450), and thus we considered the growth-non-growth inferences. Un-necessary information was removed.

- Figure 8: Too speculative, suggest removing this figure and be more careful with associated discussion.
Deleted, and subsequent figures updated. Texts updated.

- Figure 9: Too speculative, suggest removing this figure and be more careful with associated discussion.
Deleted, and subsequent figures updated. Texts updated.

- Figure S18: Too speculative, suggest removing this figure and be more careful with associated discussion.
Deleted, and corresponding texts were removed (including the previous suppl. Text no. 4)

*The discussion of the 8.2ka event is still a bit problematic, particularly as it isn't clear that this excursion in the Madagascar record is reproducible, or that it is significantly different from other isotopic excursions in the records. I would suggest further reducing the emphasis on this event, although it would be OK to still mention it as a possible link that requires further verification with additional high resolution records from this time interval. Also, in the section at line 297, please don't describe this as the 8.2ka event (this is an interpretation, not a result): instead you can say something like "A prominent isotopic excursion is evident at xx ka BP +/- xx y."
This has been considered. In the result section, we considered the Editor's suggestion about mentioning the isotopic excursion instead of mentioning directly the 8.2 ka event (see Line 264). We also revised the section 5.5 on the 8.2 event (Lines 493–516). We mentioned the need of high resolution records at that time interval at regional scale (Lines 513–516).

Minor comments:
*The sentence beginning line 59 needs to be rephrased to make sense.
Done (See Line 48–52)

*Line 303/304: I think that you are refering to "local karst conditions", not invoking "local climate conditions" as a difference between the caves. Please clarify.
Done (see Line 269)

*Section 4.3: I was expecting to see details in here describing why the mineralogy was primary and not a secondary alteration product. Consider moving this information to this section.
Good point, we moved that information in this section (see Line 289–299)

*Line 417: start a new paragraph here to break up the long section of text.

Since the texts about primary calcite were moved to earlier section, this paragraph has been revised and thus breaking up the remaining paragraph is no longer applicable.

*Line 523: I don't think "6ka event" is right in this context. I believe that this is a 6ka time slice simulation, rather than a climate event?
Revised, and we added 'although the simulation is of shorter term than the MMHI hiatus, but additional paleoclimate records are needed to improve its spatial and temporal resolution'. (Lines 463–465).

*Line 560: There is no need to bring in "Bond Cycles" here when you don't look at any other of the proposed cycles in your work.
Sentence removed. Also, the section 5.5 on the 8.2 was revised according to Editor's comments above.

*Line 629: I think you mean "migration" not "expansion" here, as this is followed by "and/or expansion".
The word is 'contraction' (i.e., expansion and/or contraction), and texts have been updated. (Lines 536).

*Figure 1b: please fix formatting so that all components of the figure are legible and not covered by labels.
Done

*Figure 1d-e: Show location within caves where samples were collected.
Done

*Figure 2: it is very difficult to compare the age information of the two speleothems in the current format of this figure. Consider combining into a single panel with two depth axes so that it is more obvious to the reader how the two age scales compare.
This figure has been revised. However, we would like to note that because of the height difference between the two samples, combining the age model into one depth series would compromise the readability of one of the models. Instead, we re-arranged the figures so that the lines of hiatus are aligned, and we added scanned images of the samples to help the reader see the three intervals in the samples. We hope this is a better version of the age model.

*Figures 5 and 6: Information about aragonite and calcite sections of the speleothems has been lost in the revisions. This is important, so please consider adding this information into these figures.
Added, and figure caption updated accordingly.

Figure 7: it would be helpful to use a symbol to show Madagascar on the maps. Is it necessary to include the Walker and the Head and Gibbard subdivisions on this figure?
Revised accordingly. Some unnecessary texts in the introductory paragraph of Sect. 5.2 that pertains to this comment were removed (Lines 364–367).

Figure 11: Add information beside these records to show information on their interpretation (e.g. double arrows showing direction of wet/dry or warm/cool, etc.)
Done, and it is now Figure 9

**Additional comments from authors:**
- The corresponding author and few other co-authors' affiliation have been updated.
- Figures S6 and S7 in the supplementary material have been updated because the corrected $\delta^{18}O$ (dashed blue lines) values have been accidentally plotted with the same axis as $\delta^{13}C$.
- The previous supplementary text no. 4 was removed and has been replaced with previous section 5.6 of the manuscript. This section can be deleted permanently if necessary.
- References in both manuscript and supplementary have been updated.